# High performance floating self-excited sliding triboelectric nanogenerator for micro mechanical energy harvesting

Li Long[1], Wenlin Liu ⬤ [1✉], Zhao Wang[1], Wencong He[1], Gui Li[1], Qian Tang[1], Hengyu Guo[1], Xianjie Pu[1], Yike Liu[1] & Chenguo Hu ⬤ [1✉]

Non-contact triboelectric nanogenerator (TENG) enabled for both high conversion efficiency and durability is appropriate to harvest random micro energy owing to the advantage of low driving force. However, the low output ($<10\,\mu C\,m^{-2}$) of non-contact TENG caused by the drastic charge decay limits its application. Here, we propose a floating self-excited sliding TENG (FSS-TENG) by a self-excited amplification between rotator and stator to achieve self-increased charge density, and the air breakdown model of non-contact TENG is given for a maximum charge density. The charge density up to $71.53\,\mu C\,m^{-2}$ is achieved, 5.46 times as that of the traditional floating TENG. Besides, the high output enables it to continuously power small electronics at $3\,m\,s^{-1}$ weak wind. This work provides an effective strategy to address the low output of floating sliding TENG, and can be easily adapted to capture the varied micro mechanical energies anywhere.

[1] Department of Applied Physics, State Key Laboratory of Power Transmission Equipment and System Security and New Technology, Chongqing Key Laboratory of Soft Condensed Matter Physics and Smart Materials, Chongqing University, Chongqing 400044, China. ✉email: liuwl@cqu.edu.cn; hucg@cqu.edu.cn

ntelligence and informatization are the current running modes of modern society, in which efficient utilization of various resources is an inevitable trend[1,2]. In terms of energy, not only the concentrated energy such as nuclear power and hydropower, etc., are needed for cities but also the distributed energy such as human motion, breeze, vibration, etc., can be harvested for powering personal/small devices[3]. Further, the broad application scenes of distributed energy have attracted attention all over the world[4]. Based on the coupling effect of triboelectrification and electrostatic induction[4,5], triboelectric nanogenerator (TENG) has been demonstrated as a more efficient energy-harvesting strategy, compared with electromagnetic generator[6,7] or piezoelectric generator[8–10] for low-frequency ambient distributed energy (5 ~ 100 times smaller than the TENG at <5 Hz working frequency). With the improvements in working modes and performance, now TENG is possibly used in biosensing[11–14], artificial intelligence[15–17], high-voltage application[18–21], and blue energy[22–25] due to its outstanding merits of low cost, simple structure, diversity of materials, flexibility, and adaptability[26,27]. In general, contact friction causes interface heat loss and abrasion, whereas the non-contact plane motion can avoid such drawbacks. Therefore, a floating mode TENG without interface contact has high durability and would have almost 100% theoretical conversion efficiency, owing to the zero friction loss[28], and can easily harvest slight motion energy, showing the greatest potential in the commercial process of TENG. It is known that the surface charge density is the key factor to enhance the output of TENG[29,30]. However, the quick decay of pre-existing charge of the non-contact mode leads to a very small output and thus limits its application[28,31–33].

To boost the charge density, various methods of chemical modification[34], contact improvement[35], environment control[36], charge pump[37], etc., are proposed and the charge density improvements from 100 to 1020 µC m$^{-2}$ are achieved for contact-separation TENG. Our proposed efficient charge excitation strategy boosts the charge density of contact-separation TENG to 1.25 mC m$^{-2}$ in air ambient[38] and further reaches 2.38 mC m$^{-2}$ by quantizing the contact status under charge excitation[39]. Recently, a ingenious design by utilizing the charge space-accumulation effect enables the sliding TENG (S-TENG) to achieve 1.68 mC m$^{-2}$ charge density[40] and a super-high charge density of 5.4 mC m$^{-2}$ is achieved by increasing the micro-electrode unit number through direct triboelectrification and charge releasing from air breakdown[41], reaching a new milestone. However, a large driving force is usually needed to drive the sliding mode TENG along with large and inevitable interface friction, or to realize the intimate contact for contact-separation TENG. Obviously, a S-TENG in contact mode is not conducive to collecting low-speed wind and wave energy even though with the introduction of interface liquid lubricant[42]. As for non-contact mode, a rotational TENG transforms contact mode at low speed to non-contact mode at high speed automatically by the centrifugal force of a built-in traction rope structure, by which TENG increases the charge in contact mode to overcome the charge decay in non-contact mode[43]. In addition, a supplementary charge method is developed to elevate output, by adding an external tribo-material to enhance the charge density of tribo-layer[44–46]. However, the enhancement of charge density only reaches 20 µC m$^{-2}$, owing to the limited triboelectrification effect. Although the charge density and output power of contact mode TENG have improved significantly, it is still a great challenge so far for the non-contact mode to improve its output electric energy. Consequently, it is desirable to improve the TENG that can achieve durability and high output performance simultaneously under a small driving force.

In this work, a floating self-excited S-TENG (FSS-TENG) is proposed with both high output performance and long-term durability by a self-excited amplification between rotator and stator, and it can be used to efficiently harvest various small mechanical energy. By introducing the unidirectional conduction voltage-multiplying circuit (VMC) and adding an excitation electrode in the non-contact rotator, the FSS-TENG itself realizes a fast exponential self-increase in charge density. To achieve maximum output charge density, the breakdown model of the non-contact TENG is given from both theory and experiments. When the structure is optimized, the FSS-TENG delivers a transferred charge of 1 µC (71.53 µC m$^{-2}$) and peak power of 34.68 mW at a speed of 300 rpm, which are 5.46 times and 3.88 times enhancement compared with the floating TENG (F-TENG) without charge excitation, respectively. Moreover, the maximum output charge density of FSS-TENG increases slightly after 100 thousand times of operation, exhibiting the ultrahigh output stability. Finally, by using wind cups as triggers, the FSS-TENG is verified to light road warning lights by harvesting wind energy at 3 m s$^{-1}$ low wind speed and to drive some small electronics continuingly. This work provides a reliable strategy to harvest random ambient energy and realize distributed energy supply.

## Results

**Structure and working principle of FSS-TENG.** Rational and efficient utilization of the new energy sources is advocated worldwide. The FSS-TENG is developed to efficiently collect wind energy with long service life. Figure 1a shows the blueprint of this work, where the FSS-TENG can be installed in coastal areas, mountainous areas, and even cities to collect wind energy to power warning lights or some sensors, to achieve self-powered micro-grid distribution and environmental monitoring. A three-dimensional structural schematic of the rotary FSS-TENG unit is shown in Fig. 1b, consisting mainly of a disk-shaped rotator and a stator. The stator has 12 fan-shaped Al electrodes with the area of about 23.3 cm$^2$ for each. The surface of the electrodes is covered with a layer of 25 µm nylon (Polyamide, PA) film, which can effectively inhibit the direct air breakdown between the metal electrodes, thus obtaining a larger charge density. As for the rotator, the six fan areas are covered with 30 µm polytetra-fluoroethylene (PTFE) film with the negative charge by using Al film friction in advance, providing the initial charge for the self-excitation process. The residual complementary areas are six Cu electrodes connected in parallel as excitation electrodes and the surface of electrodes is also covered with a layer of 25 µm PA film. There is an air gap (0.35 mm) between the stator and rotator to avoid mechanical wear. Bearing connection can reduce rotation resistance and enable the device to be easily driven, and the details of fabrication are shown in the "Methods" section. Part of the output is assigned to the Cu electrodes of the rotator by the electric brush after passing through the second-order VMC, which makes the positive charge on the Cu electrodes increase continuously, effectively improving the electrostatic induction intensity.

The simplified working schematic of FSS-TENG, as shown in Fig. 1c, is equivalent to a high-voltage source and a S-TENG in non-contact mode with a charge supplement electrode. The unidirectional diode ensures that only one type of charge is injected in the Cu electrode of the slider. The relationship between the excitation voltage and the charge in the top electrode is as follows.

$$Q = \frac{\varepsilon_0 S}{d} \cdot V_{\mathrm{E}} \qquad (1)$$

where $\varepsilon_0$ and $S$ represent the vacuum permittivity and the total area of the electrode, and $d$ is the air gap distance. Equation 1 exhibits that the charge on the top electrode is proportional to excitation voltage, showing that large excitation voltage can generate larger charge on the up electrode. The voltage source

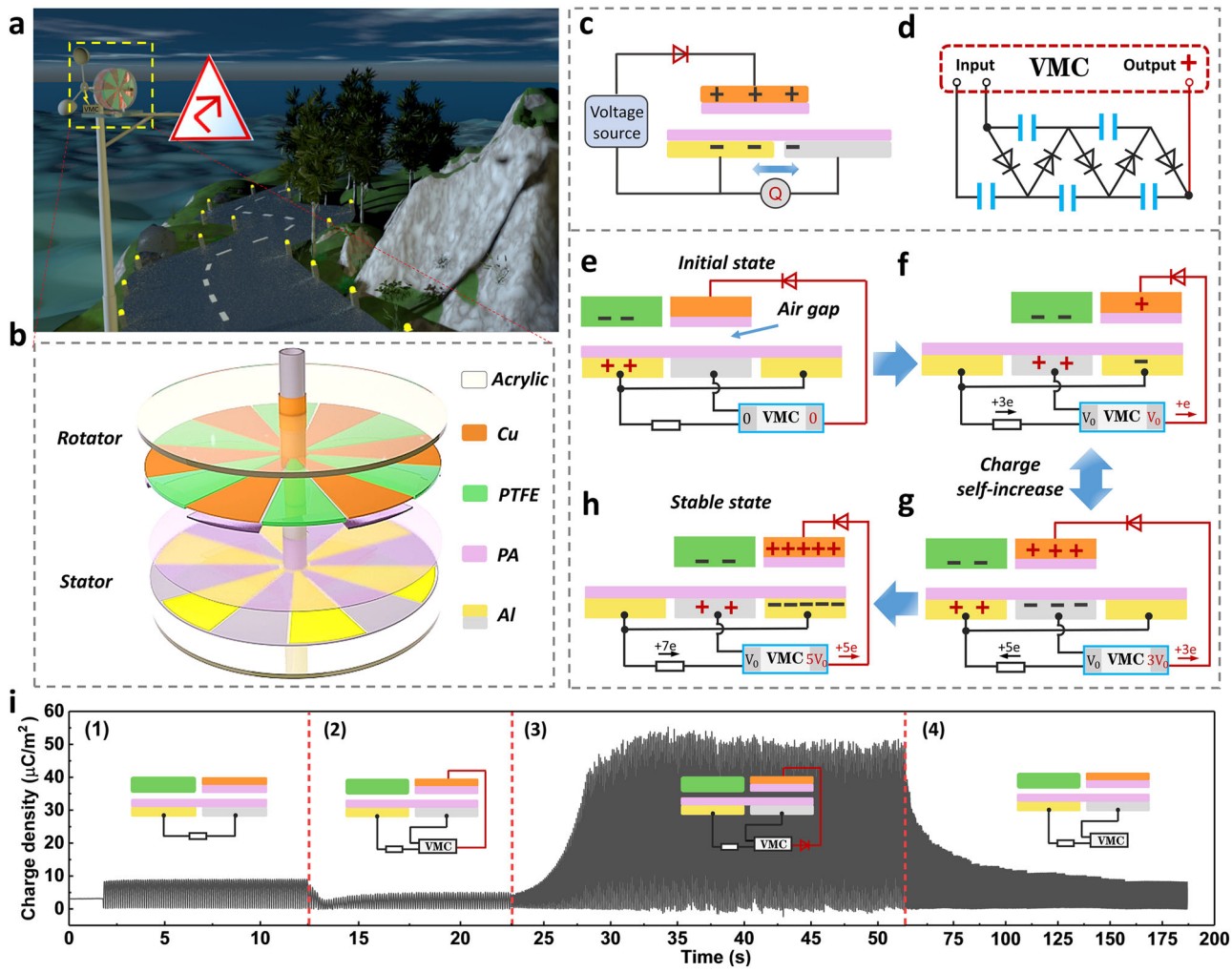

**Fig. 1 Structure and working principle of the FSS-TENG. a** Scene graph of the FSS-TENG for wind energy harvesting. **b** 3D structural schematic of the rotary FSS-TENG unit. **c** Simplified working schematic of the FSS-TENG. **d** The input/output node and scheme of the voltage-multiplying circuit (VMC). **e–h** The charge self-excitation process during periodically sliding cycle. **i** Dynamic output charge curve under four working modes, showing the powerful output boosting ability of our designed FSS-TENG.

part can be realized by a VMC, whose circuit diagram is shown in Fig. 1d. The VMC is realized through the input /output node design. The photograph of the VMC is shown in Supplementary Fig. 1, and it consists of five diodes and five ceramic capacitors. The charge self-excited process during the periodical cycle is illustrated in Fig. 1e–h. In the initial state (Fig. 1e), the PTFE has few negative charges in advance and the voltage at both ends of VMC is zero. When the rotator starts to rotate periodically, the potential difference between the Al electrodes results in AC output (Fig. 1f, g). It is noteworthy that both ends of VMC also start to be charged and the output voltage increases with the running times, and the charges assigned to the Cu electrode by the circuit (red line) also accumulate continuously. The PTFE with negative charges and the Cu electrodes with positive charges work together to force the electrons in the Al electrodes to move, enabling FSS-TENG's induction output to increase constantly. After several cycles, the voltage at both ends of VMC reaches stability and the charge injected into the Cu electrode tends to a saturation state, and the output arrives at maximum. Thus, the charge self-excited working mode is realized, as shown in Fig. 1h. Figure 1i indicates the dynamic output charge density curve under four working circumstances, including (1) the normal non-contact TENG output; (2) the output is partially assigned to VMC

but the excitation route (red line) has no diode. At this point, positive and negative electrons will be introduced into the Cu electrode at the same time, and the effect of charge accumulation cannot be realized and VMC will consume part of the energy, resulting in the output decline. (3) With a unidirectional diode, the surface charge density of Cu electrode rapidly rises to a stable value; (4) without the excitation route, the curve quickly go down to the initial value (VMC does not work), demonstrating the powerful output boosting ability of our designed FSS-TENG.

**Air breakdown model of the non-contact TENG.** For the non-contact S-TENG, the equivalent physical model is shown in Fig. 2a. The air breakdown would occur between the top electrode and the bottom electrode due to the existence of the air gap, so there is a maximum charge density ($Q_{Max}$) on the surface of the electrode. Therefore, according to the potential difference ($V_{gap}$) between the top and bottom electrodes and Paschen's law, the $Q_{Max}$ can be deduced.

There is an air gap $d$ between the plates; the capacitance of non-contact TENG can be expressed by:

$$C = \frac{\varepsilon_0 S}{d} \qquad (2)$$

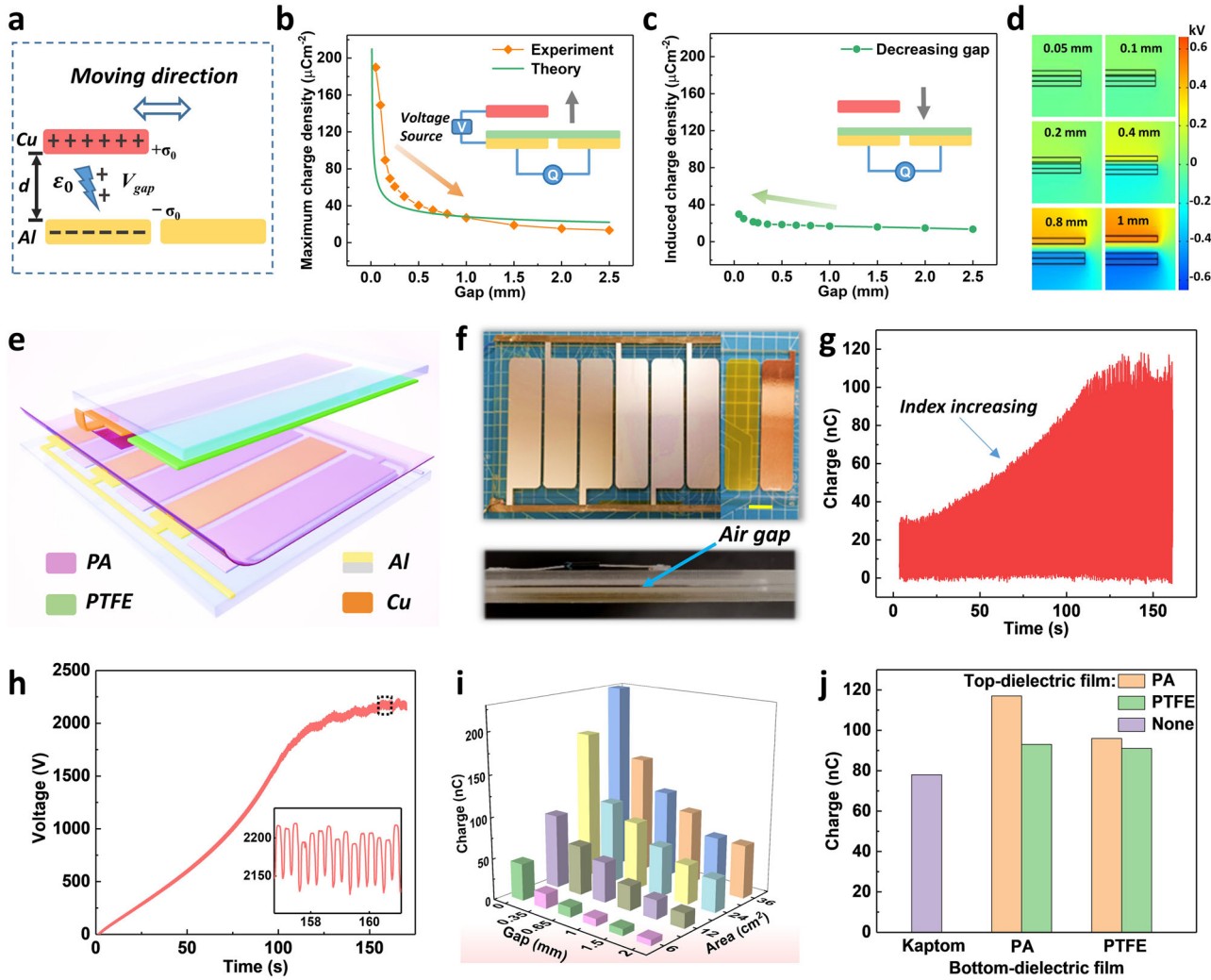

**Fig. 2 The air breakdown model of non-contact TENG and the structure influence on the output of plane FSS-TENG. a** The equivalent physical model of the non-contact TENG. **b** The plots of experimental and theoretical maximum charge density vs. the gap for the non-contact TENG with charge excitation. The inset refers to the measurement method. **c** The plot of charge density vs. the gap for the non-contact TENG without charge excitation. The inset refers to the measurement method. **d** The simulated potential distribution of difference gaps at 10 μC m$^{-2}$. **e** The 3D schematic of plane FSS-TENG. **f** Device photograph of the top view and side view. The area of each electrode and PTFE are 23.3 cm$^2$. Scale bar: 2 cm. **g** The dynamic output charge accumulation process with charge self-excitation. **h** The dynamic excitation voltage between top and bottom electrodes. **i** The output charge of plane FSS-TENG with different electrode area and air gaps (the gap of 0 mm represents the full contact state); the surface of the Al electrode is attached with a 25 μm Kapton film and the Cu electrode without film. **j** Output charge with different materials (FEP, PTFE, and Kapton, the air gap is 0.35 mm). Driving frequency is 2 Hz for **g**–**j**.

Hence, the voltage of the air gap can be expressed by:

$$V_{\mathrm{gap}} = \frac{Q_{\mathrm{Max}}}{C} = \frac{dQ_{\mathrm{Max}}}{\varepsilon_0 S} \qquad (3)$$

Obey Paschen's law, the voltage that causes air breakdown between two parallel plates follows:

$$V_{a-b} = \frac{A(Pd)}{\ln(Pd) + B} \qquad (4)$$

where $P$ is the pressure of the gas and $d$ is the gap distance between two parallel plates. $A$ and $B$ are the constants determined by the composition and the pressure of the gas. For the air at standard atmospheric pressure of 101 kPa, $A$ is $2.87 \times 10^5$ V (atm m)$^{-1}$ and $B$ is 12.6.

To avoid the air breakdown effect, the $V_{\mathrm{gap}}$ needs to be smaller than $V_{a-b}$. Thus, the following relationship is needed:

$$V_{\mathrm{gap}} \leq V_{a-b} \qquad (5)$$

According to the Eqs. 2–4, the $Q_{\mathrm{Max}}$ of the electrode can be expressed by:

$$Q_{\mathrm{Max}} \leq \frac{AP\varepsilon_0 S}{\ln(Pd) + B} \qquad (6)$$

Obviously, the maximum charge density of non-contact TENG decreases with the increase in the air gap. The theoretical calculation is plotted in Fig. 2b. In the measurement process, we provide a maximum charge density to the electrode by applying a high voltage source, the diagram of the test is shown in the inset and a PA film is affixed to the bottom electrode to protect the instrument by avoiding direct electric breakdown between the top and bottom electrodes. It can be seen from the results that the maximum charge density decreases with the increase in the gap,

which is in good agreement with the trend of theoretical calculations considering the external influence of the actual environment and the charge of induced charge at different air gaps. Therefore, for the non-contact TENG, the air breakdown in the gap has a great influence on the maximum charge density of TENG.

On the other hand, the secondary influence of induced charge at different gaps is tested, as shown in Fig. 2c, where no high voltage source is applied to supplement the charge. The induced output charge density only increases slightly as the gap decreases, indicating only a slight influence of the gap change. The simulated results by Comsol Multiphysics in Fig. 2d display the field strength forming between the top and the bottom electrode with different gaps at a charge density of $10 \mu C m^{-2}$. When the gap is gradually reduced, the electric field intensity changes within 1% each 0.1 mm, so the induced charge does not change much and does not show the trend of multiple fold increase. Thus, the induced charge caused by the gap change is not the main influence on the maximum charge density. It is concluded that the maximum charge density for the non-contact TENG is mainly affected by air breakdown and declines with the increase in the air gap, which is different from the contact-separation TENG determined by the thickness of the dielectric film. To a certain extent, the gap is equivalent to a layer of the dielectric film.

**Optimization of structural parameters**. To better investigate the structure and material influence on the FSS-TENG, a planar structure was used. The three-dimensional structure diagram is shown in Fig. 2e, which mainly includes two parts, stator and slider, and each electrode has a size of $23 cm^2$. The details of the fabrication are shown in the "Methods" section. The top of Fig. 2f exhibits the top view of the device and the bottom is the side view. It is obvious that there is an air gap between the slider and the stator. A programmable linear motor is used as the driving force of the slider to operate in floating mode at a frequency of 2 Hz. Supplementary Fig. 2 shows the initial transfer charge of about 42 nC without self-excitation. After being connected with the self-excitation circuit, the output charge and excitation voltage between the top and bottom electrodes exponentially increase and stabilize to 118 nC and 2128 V, respectively, as shown in Fig. 2g, h. To further increase the output of TENG, multiple pairs of sliders (a PTFE and an electrode in one pair) are used and the induction electrodes are connected in parallel, and the schematic diagram of the slider with two pairs of excitation electrodes reaching a stable state is shown in Supplementary Fig. 3. Supplementary Fig. 4 shows that the transferred charge and excitation voltage depends on pairs of sliders, respectively. Their initial charges are shown in Supplementary Fig. 5. It is clear that the increase in the excited charge is proportional to the pairs, but the excitation voltage finally tends to the same value, because the value is determined by the order of VMC, and the pairs only affect the speed of voltage growth. When the slider pair increases from one to three, the more induced charge can be supplied in one cycle, by which the excitation voltage takes less time for 130 s, 68 s, and 48 s to reach stability at a frequency of 2 Hz, respectively. Although the excitation voltage curve for three pairs seems to be almost similar to that for two pairs due to the slight difference of initial states (the initial state has a huge influence on the waveform of output curve), the excitation time is still shorter with three electrodes at the last. Besides, considering the integration of multiple slider pairs and the deformation of the device, the different length–width ratios of electrode needs to be optimized. Under an air gap of 1 mm, the charge output increment has an obvious difference (Supplementary Fig. 6). If multiple

electrodes are integrated, considering the whole structure of the device, the length–width ratio about 3/8 is chosen for the following tests. The air gap is an important parameter affecting the induction output of non-contact TENG. The output charge of the plane FSS-TENG with different electrode areas of the same length–width ratio and different air gap from 0 to 2 mm are presented in Fig. 2i. The transfer charge increases with the decrease in the air gap and with the increase in area, which is consistent with the previous work[28,33]. Further, their initial charge of the electrode is shown in Supplementary Fig. 7. To avoid air breakdown, covering a layer of the dielectric film on the surface of the top and bottom electrodes is an effective method. Three kinds of dielectric film (positive : PA > Kapton > PTFE : negative) with different abilities for fettering charge are chosen for the test[47]. Figure 2j displays the transfer charge with different materials under an air gap of 0.35 mm. It reveals that the transfer charge is largest with PA film on the surface of both stator and slider electrodes. As the charge on the PA film is easy to dissipate and does not accumulate on the film surface to shield the charge on the electrode, a stable and highly effective output is achieved. However, Kapton and PTFE are both electrets; they have strong electronegativity and a strong ability to hold electrons, which can easily form an electrostatic shielding between induction electrodes and decrease the output charge. The output of TENG with dielectric film (PA or PTFE) attached to the top electrode is larger than that of none, which further indicates that the dielectric film can suppress the direct air breakdown between the metal electrodes to obtain a larger surface charge density. Their initial charge provided each time is almost similar and detailed dynamic excitation processes are shown in Supplementary Figs. 8 and 9. In addition, the influence of the capacitance value in VMC on excitation charge is also tested[48]. To have a faster excitation speed, five small capacitances are chosen for comparison as shown in Supplementary Fig. 10, from which we find that as the capacitance increases, the excitation time also increases. In sufficient time, the second-order VMC voltage with different capacitances can reach stable 2000 V and this excitation voltage can enable the excitation electrode to reach the maximum charge. Therefore, in the whole testing process, the second-order VMC with 2.2 nF capacitance is used. The effect of different temperature and humidity on the surface charge density of FSS-TENG are shown in Supplementary Fig. 11. The output shows a slight downward trend with the increase of temperature and humidity, as the electron thermionic emission effect of induced charges on the surface of the electrode increases when the temperature rises and more water molecules take away the charges on the electrode surface in high-humidity air, which reduces charge density and output performance of FSS-TENG. When the FSS-TENG works under the illumination with the simulated solar light in the standard light intensity of $100 mW cm^{-2}$ for half an hour, the output charge of FSS-TENG decreases slightly, which is mainly caused by the increase of device temperature during the irradiation process (Supplementary Fig. 12). Thus, the temperature is controlled within 15–35 °C and humidity about 35–55% RH (Relative Humidity) of the experimental test environment.

**Performance of FSS-TENG**. Based on the parameter optimization of the planar structure, the rotary FSS-TENG device is designed, as shown in Fig. 3a, which contains six excitation electrodes in parallel to increase the output. Figure 3b–d show the dynamic output of charge, current, and load voltage (10 MΩ) with charge self-excitation at 300 rpm., which reaches 1 μC, 76 μA, and 470 V just in 5 s, respectively. Correspondingly, the surface charge density of the Cu electrode reaches $71.53 \mu C m^{-2}$ in the floating state. The measured output performances of

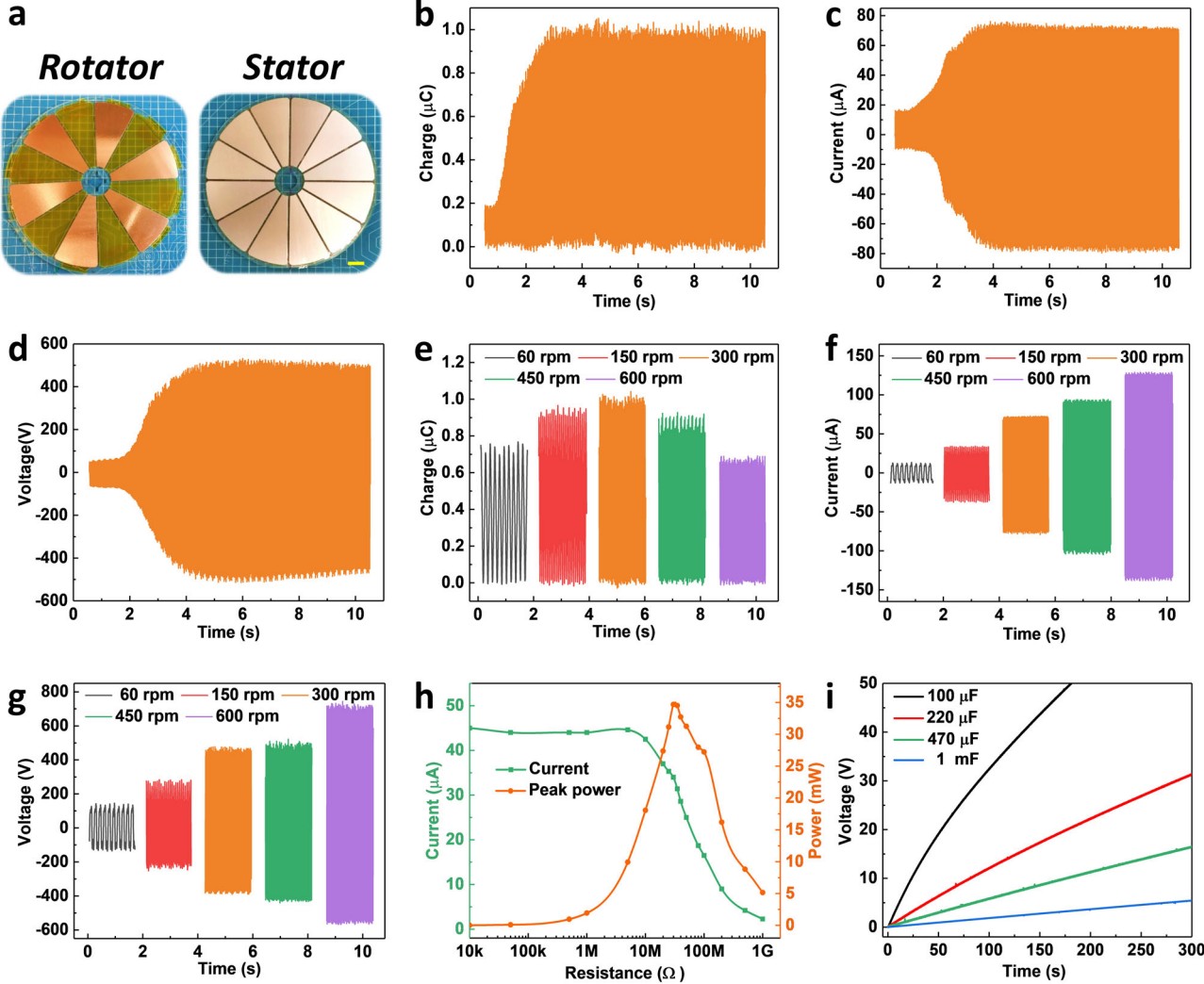

**Fig. 3 Performance of rotary FSS-TENG. a** Device photographs of stator and rotator. Scale bar: 2 cm. **b** Dynamic charge, **c** current, and **d** voltage output with charge self-excitation, at 300 rpm, respectively. **e** Transferred charge, **f** current, and **g** voltage at different rotational speeds. **h** Matching impedence and output power at 300 rpm **i** Voltage curves of charging capacitors at 300 rpm.

FSS-TENG at different rotational speeds are presented in Fig. 3e–g. The output current and load voltage increase with the increase in rotating speed. However, the transferred charge tends to increase first and then decrease, which may be that the charge excitation does not reach saturation in a short time at lower rotational speeds, whereas higher rotational speed affects the gap between the electrodes. When the speed increases, the centrifugal force and vertical tremor of the rotator increase correspondingly due to the imperfect fabrication craft, causing a increase in the gap between the rotator and the stator. Thus, the maximum output charge declines with the increase in speed. In addition, the power of FSS-TENG at 300 rpm with varied external load resistance is shown in Fig. 3h, where the peak power of 34.68 mW is obtained at 30 MΩ. The transferred charge, current, and peak power of F-TENG without charge excitation as shown in Supplementary Fig. 13, are 183 nC, 15.8 μA, and 8.94 mW at 300 rpm, respectively (Supplementary Fig. 13b–d). The transferred charge and the peak power of the FSS-TENG are 5.46 times and 3.88 times as much as that of the F-TENG respectively, displaying a power enhancement of this design. The comparison of charge density with the reported works for the non-contact TENG is shown in Supplementary Table 1. Then the capabilities of the TENG for charging different capacitors (Fig. 3i)

demonstrate that the voltage of the capacitor of 100 μF and 1 mF can be charged to 50 V within 181 s and 5.43 V within 300 s, respectively. However, the capacitor of 100 μF and 1 mF is charged to 14.74 V within 181 s and 1.60 V within 300 s by the F-TENG, as shown in Supplementary Fig. 14. Compared with the F-TENG, the charging rate of the FSS-TENG is improved by 3 ~ 4 times in the same process.

It is reported that charge density on PTFE could be up to $5 \times 10^{-4}\,C\,m^{-2}$ with a theoretical lifetime of several hundred years[49]. To determine the stability of initial charge on PTFE, the output charge of F-TENG is tested as shown in Supplementary Fig. 15, in which the output charge decreases slightly (from 16.98 to 14.66 μC m$^{-2}$) after intermittent testing of 15 days (the ambient humidity is controlled at 40–50%), proving the high stability of the charge on PTFE. PTFE is a class of electret, which has a strong ability to retain surface charge, so it can provide a low initial charge to the FSS-TENG for a long time.

The degradation of output performance caused by the abrasion of devices is a problem to be solved for a long time. The structural design of this work has greatly improved the output of TENG. Figure 4a compares the output charge trend of the S-TENG with the FSS-TENG during 100 thousand times operation, which reveals that the output of S-TENG declines to 34% due to wear

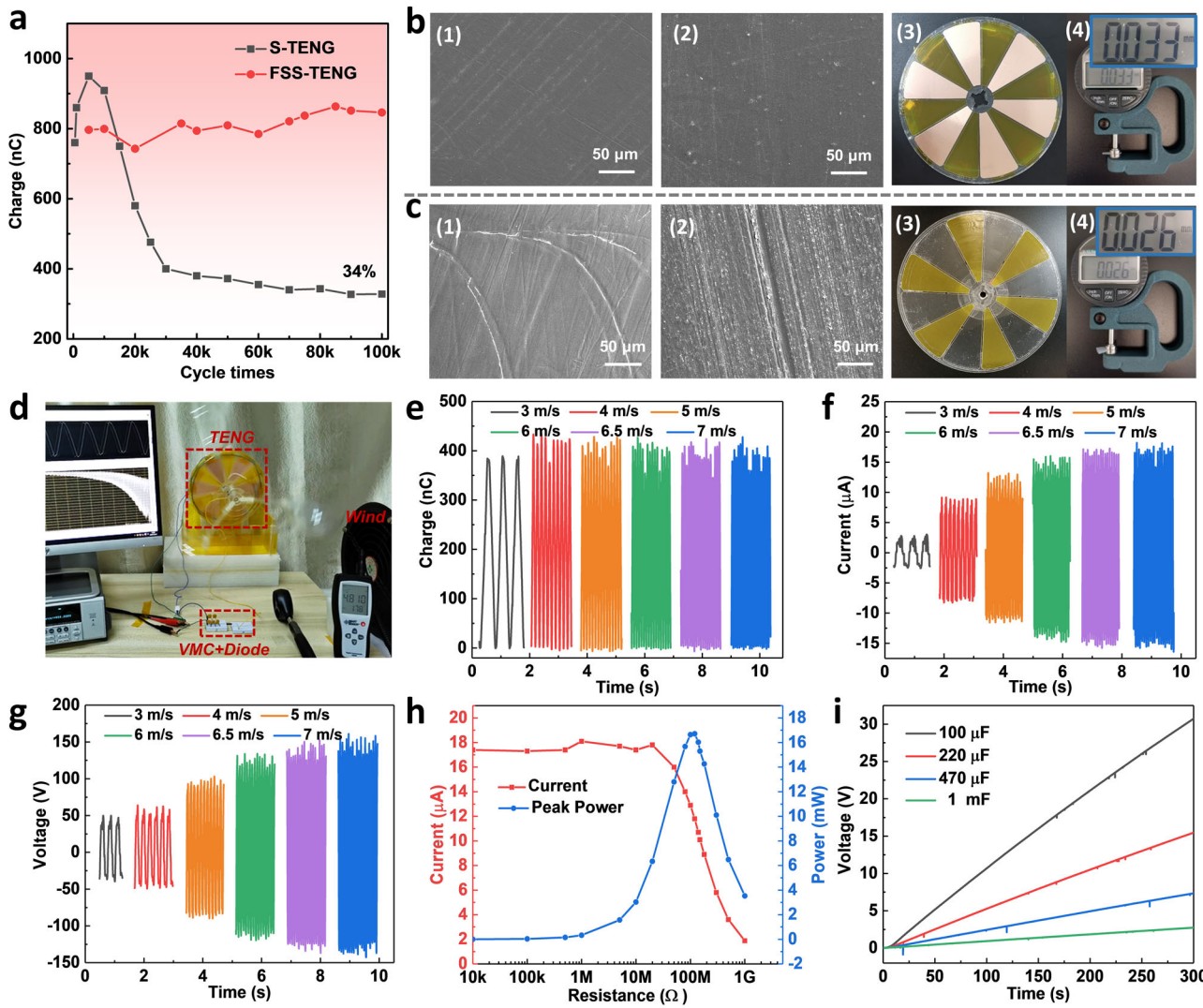

**Fig. 4 Durability and output performance of harvesting wind energy. a** Durability test of the FSS-TENG and sliding TENG (S-TENG) with the operating for 100 thousand cycles. Abrasion of the surfaces for the rotator of **b** FSS-TENG and **c** S-TENG, respectively, indicating the super long-term reliability of FSS-TENG. SEM images of (1) PTFE and (**2**) PA. (3) The device abrasion photographs of rotator part. (4) The thickness of PTFE after stability test. **d** Scene photograph of harvesting wind energy by the FSS-TENG. **e** The transferred charge, **f** current, and **g** voltage of FSS-TENG at different wind speeds. **h** Matching impedence and output power at 7 m s$^{-1}$ wind speed. **i** Voltage curves of capacitors charging at 7 m s$^{-1}$ wind speed.

and high heat after long time work, but the output of FSS-TENG is relatively stable with a slight increase, indicating the super long-term reliability. Parts of the curves for the first and the last hour are shown in Supplementary Fig. 16. Comparison of the scanning electron microscopic (SEM) images of PTFE and PA after 100 thousand cycles, as shown in Fig. 4b (FSS-TENG) and Fig. 4c (S-TENG), with the SEM images of unused materials (Supplementary Fig. 17) clearly shows that the wear of contact friction is serious and the zero material loss of non-contact mode. Figure 4b, c3 are the rotator photographs and Fig. 4b, c4 are the thickness of PTFE after 100 thousand cycles. It can be seen that the rotator surface of S-TENG has much debris and the PTFE film even becomes thinner under such serious interface abrasion. The photograph of the stator for S-TENG after 100 thousand cycles is shown in Supplementary Fig. 18, exhibiting the similar high abrasion of the stator.

To demonstrate the commercial potential of FSS-TENG, a home-made wind cups as triggers is used to collect wind energy. Figure 4d shows the scene photograph. The dynamic process of charge excitation for the FSS-TENG driven by wind is shown in

Supplementary Movie 1. The results are plotted in Fig. 4e–g under different wind speeds from 3 to 7 m s$^{-1}$, from which we can see that the transferred charge remains about 420 nC, different from the trend driven by a motor, mainly because the corresponding rotational speed is still relatively low even at high wind speed (19~140 rpm) (Supplementary Table 2). Besides, the uneven force and large fluctuation of wind energy easily lead to large vertical tremors and decrease the transfer charge at low speed. Consequently, the output charge of FSS-TENG shows a constant tendency due to the offset of the above two factors when driven by wind at last, although the short current and load voltage (10 MΩ) increase with the increase in wind speed, arriving at 17.6 μA and 150 V under 7 m s$^{-1}$, respectively. We believe that the FSS-TENG could be driven easily at <1 m s$^{-1}$ wind speed with the further optimization of the manufacturing process. It is noteworthy that the output is smaller than measured by the motor, owing to that the rotator is not stable in rotation and the air gap is larger in using wind power, and it can be optimized in the assembly process to improve the output. At the resistance of 120 MΩ, the FSS-TENG achieves the peak power of 16.7 mW under 7 m s$^{-1}$ wind

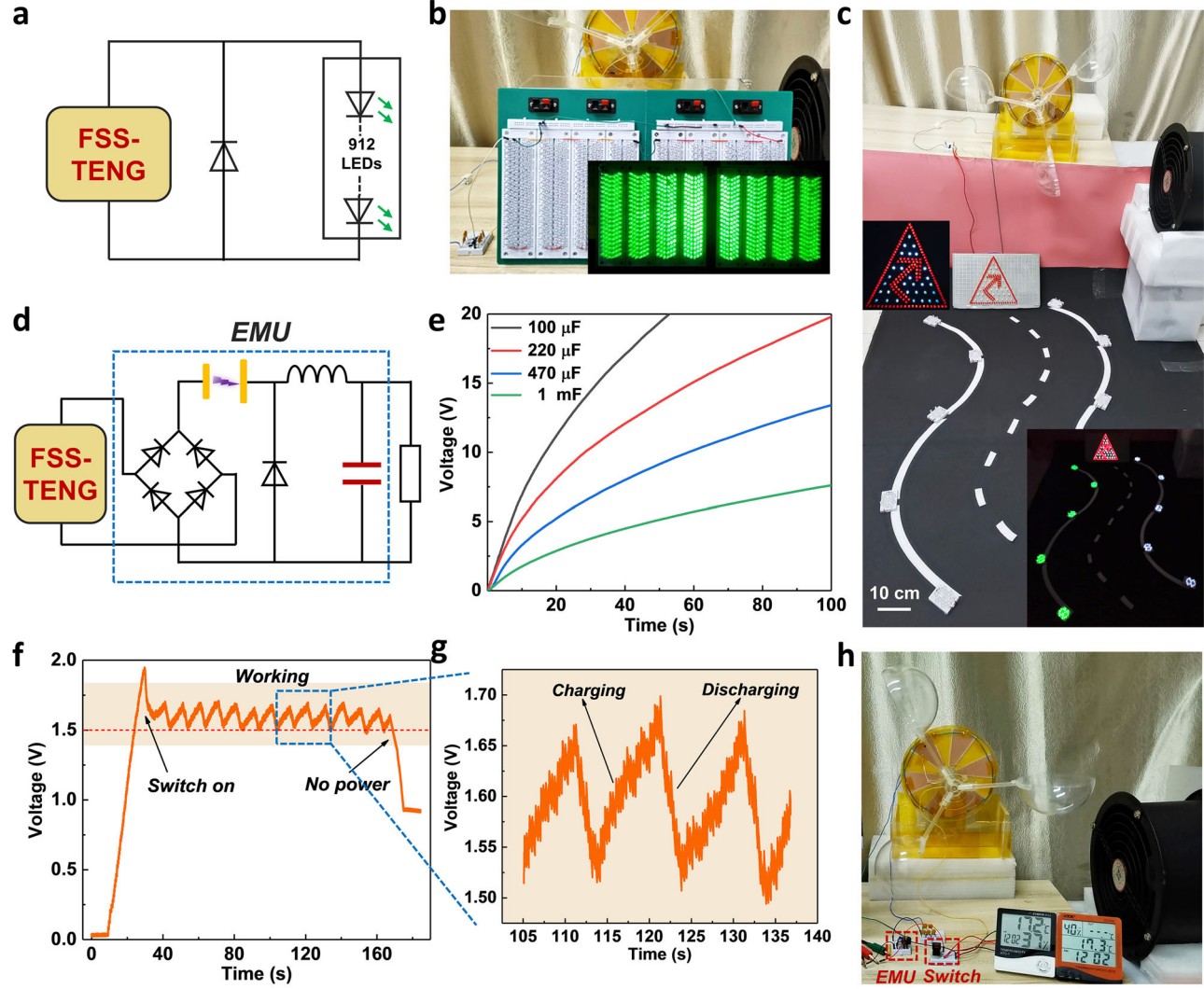

**Fig. 5 Demonstration of the FSS-TENG to drive devices. a** Circuit diagram and **b** real photographs of the FSS-TENG for lighting 912 LEDs. **c** Directly driving the simulated road warning lights with 348 LEDs at a wind speed of 5 m s$^{-1}$ (all of the lights have a diameter of 5 mm). **d** Circuit diagram of the FSS-TENG with an energy management unit (EMU) for powering electronic equipment. **e** Voltage curves of capacitors charging at 5 m s$^{-1}$ wind speed with EMU. **f** Voltage–time curve of two temperature hygrometers in parallel at 3 m s$^{-1}$ wind speed with the EMU. **g** The detailed voltage curve of the selected area in **f**. **h** The real photograph of the temperature hygrometers powered by the FSS-TENG.

speed, as shown in Fig. 4h. The voltage curves of four different capacitors are depicted in Fig. 4i, where the capacitors of 100 μF, 220 μF, 470 μF, and 1 mF are correspondingly charged to 30 V, 15.5 V, 7.3 V, and 2.7 V in 300 s.

**Application demonstration of FSS-TENG**. To provide a more intuitive demonstration of FSS-TENG's ability to effectively convert wind energy, the output is rectified with a diode to light 912 green light-emitting diodes (LEDs) in series at 5 m s$^{-1}$ wind speed (Supplementary Movie 2) and its circuit diagram and the real photographs are shown in Fig. 5a, b. Figure 5c exhibits the FSS-TENG used in simulation scenarios of road warning lights (Supplementary Movie 3). With the help of the energy management unit, which consists of a spark switch with a working voltage of 400 V, an inductance of 20 mH, a input capacitor of 110 pF, and a output capacitor of 470 μF, as presented in Fig. 5d, the charging process for capacity by the FSS-TENG is greatly improved. The capacitors of 220 μF, 470 μF, and 1 mF are charged to 19.7 V, 13.4 V, and 7.6 V in 100 s, specifically, the 100 μF capacitor is charged to 20 V in 52 s at 5 m s$^{-1}$ wind speed (Fig. 5e). The charging rate is improved by about four times. At a

low wind speed of 3 m s$^{-1}$, it effectively drives two temperature hygrometers in parallel. Figure 5f plots the voltage–time curve in working, and Fig. 5g is the detailed voltage curve of the selected area (Supplementary Movie 4), which proves that the FSS-TENG can supply power for practical applications continuingly. Figure 5h is the real photograph. The above have strongly demonstrated the FSS-TENG's ability to capture wind energy and its great potential in the area of microgrids.

## Discussion
In summary, the air breakdown model of non-contact TENG is given to achieve its maximum charge density. In addition, the FSS-TENG has been designed to achieve a high power output and high durability for effectively collecting micro mechanical energy. The self-excitation achieved by using VMC enables the F-TENG transferred charge to rise to 1 μC within 5 s at 300 rpm, with a surface charge density of 71.5 μC m$^{-2}$, which is 5.46 times of the F-TENG, demonstrating its powerful output. Importantly, after 100 thousand times of stability testing, the FSS-TENG exhibits excellent durability without any damping in output performance and any abrasion in the device. Furthermore, the FSS-TENG is

applied to collect wind energy and to light 912 LEDs directly, which can be applied to road warning lights. With energy management, the FSS-TENG can continuously power two parallel temperature hygrometers at $3\,m\,s^{-1}$ wind speed. This work provides a promising strategy for a long-term solution to the future energy demand of microgrids or even large grids.

## Methods

**Fabrication of the plane-type FSS-TENG.** Stator: (1) Using laser cutter to cut the 4 mm acrylic plate into the corresponding size as the base. (2) Three pairs of chamfered Al electrodes with a 2 mm distance between them were pasted on the base. The electrodes were connected with wires for the measurement of output performance. (3) A dielectric film (PA/PTFE/Kapton) with the same size as the base was attached to the Al electrode to prevent direct air breakdown between metal electrodes. In the covering process, a very thin layer of Epoxy glue was coated on the surface of the Cu electrode by scraping film approach first, then the PA film was covered on the epoxy glue, and in the last the PA film was pasted on the surface of Cu electrode by epoxy glue. Especially, the thickness of the epoxy glue layer could be neglected due to its very thin thickness (<1 μm).

Slider: (1) Using laser cutter to cut the 4 mm acrylic plate into a smaller size as a base for the slider. (2) A 30 μm thickness PTFE film rubbing with Al film in advance was attached to the base, to provide the initial induced charge. A piece of 25 μm thickness Cu electrode with the same size as Al electrode was affixed at the adjacent interval of 2 mm and then covering a layer of 25 μm thickness dielectric film (PA/PTFE/Kapton) on the surface to avoid the air breakdown. (4) Assembly: Acrylic sheets (2 mm × 4 mm) with different thicknesses (0.35 mm, 0.65 mm, 1 mm, 1.5 mm, 2 mm) were used as spacers to form a certain gap between the stator and the slider to achieve a non-contact mode. A part of the output was used to realize charge accumulation by connecting the Cu electrode of the slider through a self-VMC.

**Fabrication of the rotatory FSS-TENG.** Stator: (1) An acrylic plate with a thickness of 4 mm was cut into a disk with an inner diameter of 1.9 cm and an outer diameter of 21 cm as the base using a laser cutter. A circular hole in the middle was used to place the bearing. (2) Using a laser cutter, a set of complementary fan-shaped-array electrodes shallow ditches with an inner diameter of 7 cm, an outer diameter of 20 cm, and a radial angle of about 30° were printed on the base surface. Al electrodes were affixed with an area of about 23.3 cm² and the distance between electrodes was 2 mm. The two sets of electrodes were respectively connected with a wire for the output measurement. (3) A 25 μm PA film was applied to the surface of Al electrode. The covering process was the same as above.

Rotator: (1) Use a laser cutter to cut a thickness of 4 mm acrylic plate with the same size of stator as the base and half of them were coated with a Cu film of 25 μm thickness, and a 25 μm PA film was applied to their surface. (2) PTFE flim with a thickness of 30 μm was affixed to the rest area to provide the initial induced charge. (3) The stator and rotator were separated by Acrylic sheets to form an air gap of 0.35 mm, the shaft was connected to the bearings mounted on the stator and rotator, and the bearings were treated by lubricating oil. (4) A brush was used to connect the Cu electrode of the rotator to collect the output.

**Electrical measurement and characterization.** The slider of plane-type FSS-TENG was driven by a linear motor (LinMot S01-72/500). A commercial programmable Hybrid Servo Drive (TC5510) was applied to drive the rotary of rotation-type FSS-TENG. An adjustable wind speed turbine system controlled by applied voltage was established to simulate the wind in the nature. The short-circuit current and the transferred charge of the TENG were tested by an electrometer (Keithley 6514), and the load voltage was measured by a high-speed electrostatic voltmeter (Trek model 370). A simulated solar light in the standard light intensity of 100 mW cm⁻² was used. The temperature was controlled within 15–35 °C and humidity about 35–55% RH.

## Data availability

Data are available from corresponding authors W.L. (liuwl@cqu.edu.cn) and C.H. (hucg@cqu.edu.cn) upon reasonable request. Source data are provided with this paper.

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

## Acknowledgements

This work was supported by the Fundamental Research Funds for the Central Universities (2019CDXZWL001), National Natural Science Foundation of China (52073037, 51772036, 62004017, and 51902035), and Chongqing graduate tutor team construction project (ydstd1832).

## Author contributions

C.H., W.L., and L.L. provided the design for the experiment and the implementation steps. L.L. fabricated the device and measured the output performance. L.L., W.L., Z.W., and W.H. assisted in plotting and analyzing data. L.L., W.L., and C.H. wrote the manuscript. G.L., Q.T., H.G., X.P., and Y.L. provided some advice. All authors contributed to the manuscript.

## Competing interests

The authors declare no competing interests.
