## [Peer Review File · Nature Communications]

REVIEWER COMMENTS

Reviewer #1 (Remarks to the Author):

The manuscript "High performance floating self-excited sliding triboelectric nanogenerator for micro mechanical energy harvesting", by Long et al. achieved a high charge density in non-contact triboelectric nanogenerators using a self-exciting method. This achievement opens the door for the development of highly durable TENG devices with good output performances and is thus of current interest, being a timely report in this rapidly evolving field. It is therefore my view that this manuscript has the interest to be published in Nature Communications, although major revisions should be performed.

-The PTFE film is said to be negatively charged in advance. The authors should mention how this process was performed. Being a non-contact mode TENG, what is the stability of these charges in the floating TENG? If the sliding process is stopped do these pre-existing charges decay and in what time scale?

-The air gap between the stator and rotator is of 0.35 mm. How was such small gap obtained and maintained constant? Does this distance accounts already for the thicknesses of the triboelectric/electrode materials? In Fig. 2i, the air gap was varied from 0 to 2 mm. Is 0 mm a contact-mode TENG?

-The authors state that "The surface of the electrodes is covered with a layer of 25 μm PA film", but do not mention the process used in this covering step. PA should also be defined in the text. Why was PA chosen?

-What are the VMC components for the experiments performed in section 2.1? How does the overall performance change with these components?

-How was Q_{max} measured experimentally and how were the different gaps obtained and measured?

-In Fig. S3 the excitation voltage seems to behave almost similarly already for 2 and 3 pairs. What is the reason for that?

-With increasing rotational speed, the transferred charge eventually starts to decrease. The authors claim that "higher rotational speeds may affect the induction between the electrodes", but it is not clear how this can occur.

-The charging of the different capacitors using the self-exciting TENG is not compared with the same process with the non-exciting case. What is the charging improvement with the proposed concept?

-The EMU presented in Fig. 5d should be clearly detailed.

-Recent reviews on biomedical (Recent advances of triboelectric nanogenerator based applications in biomedical systems; <https://doi.org/10.1002/eom2.12049>), human-machine (Human-Machine Interfacing Enabled by Triboelectric Nanogenerators and Tribotronics; <https://doi.org/10.1002/admt.201800487>), high voltage (High-voltage applications of the triboelectric nanogenerator—Opportunities brought by the unique energy technology; <https://doi.org/10.1557/mre.2020.2>) and blue energy (Emerging triboelectric nanogenerators for ocean wave energy harvesting: state of the art and future perspectives; <https://doi.org/10.1039/D0EE01258K> and Triboelectric nanogenerators for a macro-scale blue energy harvesting and self-powered marine environmental monitoring system; <https://doi.org/10.1039/C9SE01184F>) triboelectric nanogenerators have been published and should be cited in the manuscript.

-There is a small number of typos in the manuscript that should be corrected (e.g. "Tow" in Fig. S3).

Reviewer #2 (Remarks to the Author):

This manuscript presents the floating self-excited sliding TENG (FSS-TENG) having high charge density based on air breakdown model. The non-contact TENG has been demonstrated to overcome the durability and efficiency issue of traditional TENG, however, charge density and output power should be improved. The author proposed the air breakdown model to calculate the optimal structure for higher charge density. It is remarkable that the FSS-TENG first achieved the first charge density of $71.53 \mu\text{C}/\text{m}^2$, which is 5 times compared to traditional floating TENG. However, the reviewer thinks this manuscript needs additional data supplement to be published in Nature Communications after minor revision. The required revision data are summarized as follows.

1. Regarding the transfer charge (Fig. 2j), the material properties-based explanation is recommended for clear understanding. The manuscript compared theoretically for different electrode gap and area only. The reviewer suggest that the material analysis of Kapton, PA, and PTFE should be included.
2. In addition, the detail explanation should be required for comparing the charge depending on rpm and wind speed (Fig. 3e and 4e). Although the very high rpm affected the induction to lower the transferred charge, high wind speed didn't induce low charge.
3. The quality of some figures should be improved for better understanding. So below papers are recommended to be referred to enhance the informativeness of figures.
 - A. "Stretchable piezoelectric nanocomposite generator." Nano Convergence, 3, 12, 2016
 - B. "Self-assembled incorporation of modulated block copolymer nanostructures in phase-change memory for switching power reduction." Acs Nano, 7(3), 2651, 2013

Reviewer #3 (Remarks to the Author):

The paper draft is interesting, and a novel device is proposed.

There are several notable grammatical issues that should be resolved prior to publication. Specifically, the article is missing the article "the" throughout and this oversight makes the draft difficult to read.

In addition, the authors give several statements such as, " The air breakdown model of non-contact TENG is first given to achieve a maximum charge density output," where the article is missing and the way in which "first" is used gives the implication that there is a second item to be described in the section or paragraph. However, the reviewer believes the authors intend these sentences to reflect that this work/device is described "for the first time". Hence, sentences like this in the draft should be modified for clarity.

There are also a few instances in the Introduction that could be improved. For example, for the sentence that begins with "Based on the" in lines 51 – 55, the authors state that the TENG is more efficient than an electromagnetic generator and piezoelectric generator, however, no actual numbers are provided for comparison. Indeed, the authors do provide two references for this sentence. However, the provision of actual numbers in the parenthesis in this sentence would strengthen the claim.

In sentence, 61, the authors state that "contact has high durability and almost 100% high conversion efficiency." This claim would be enhanced if the authors report the actual value versus a rounded number as it is clear that 100% efficiency is not possible.

The sentence on line 167 is missing a verb.

The authors provide an equation for voltage between two parallel plates and maximum charge density that includes variables, P , A , and B , which are functions of composition and pressure of the air. The authors assume and validate this model using COMSOL assuming that the air is under these conditions. However, wind turbines are many meters above sea level and therefore, the air has different characteristics than those assumed in the model. This paper would be enhanced if the theoretical and numerical models took more realistic parameters for this application into account. In doing this, others using the results would have a better understanding of the advantages and limitations of the device.

In the Optimization of Structural parameters section, the authors describe how the various parameters such as the number of electrodes are varied, but an approach to the optimization strategy, approach, or analytical model associated with the things such as the number of electrodes is not described in a meaningful way. Could the approach to the optimization methods be described in a generalized fashion? Can this approach be mathematically modeled?

Point-by-Point Response to the Reviewer's Comments

(Comments in black, response in blue)

Reviewer #1 (Remarks to the Author):

The manuscript “High performance floating self-excited sliding triboelectric nanogenerator for micro mechanical energy harvesting”, by Long et al. achieved a high charge density in non-contact triboelectric nanogenerators using a self-exciting method. This achievement opens the door for the development of highly durable TENG devices with good output performances and is thus of current interest, being a timely report in this rapidly evolving field. It is therefore my view that this manuscript has the interest to be published in Nature Communications, although major revisions should be performed.

Response: We highly appreciate the reviewer's positive comments on our work as “This achievement opens the door for the development of highly durable TENG devices”. And we have revised the article carefully.

Comment 1:

The PTFE film is said to be negatively charged in advance. The authors should mention how this process was performed. Being a non-contact mode TENG, what is the stability of these charges in the floating TENG? If the sliding process is stopped do these pre-existing charges decay and in what time scale?

Response: Thank the reviewer for raising this valuable question. During the experiment, we used a piece of Al film to rub with PTFE film three or four times, and the PTFE film is negatively charged in advance by above triboelectrification.

PTFE belongs to electret materials, which can maintain a permanent static charge in its surface on condition of no environmental impacts (*J. Mater. Chem. A* **2014**, 2, 2079-2087). However, the charged ion or water molecule could decrease the surface charge of PTFE in the actual situation. Hence, we test the charge-time curve in **Supplementary Fig. 13** to display its stability, and the result shows that the surface charge of PTFE can maintain over 86.34% of its initial value (The corresponding surface charge density is

14.66 $\mu\text{C m}^{-2}$) after intermittent testing of 15 days, indicating long-term stability. We have added a corresponding description in the manuscript to clarify this point.

Supplementary Fig. 13 Stability test of F-TENG.

It is reported that charge density on PTFE could be up to $5 \times 10^{-4} \text{ C m}^{-2}$ with a theoretical lifetime of several hundred years⁵⁰. To determine the stability of initial charge on PTFE, the output charge of F-TENG is tested as shown in *Supplementary Fig. 13*, in which, the output charge decreases slightly (From 16.98 to 14.66 $\mu\text{C m}^{-2}$) after intermittent test for 15 days (the ambient humidity is controlled at 40% to 50%), proving the high stability of the charge on PTFE. PTFE is a class of electret, which has an extremely strong ability to retain surface charge, so it can provide the initial charge to FSS-TENG for a long time.

Comment 2:

The air gap between the stator and rotator is of 0.35 mm. How was such small gap obtained and maintained constant? Does this distance accounts already for the thicknesses of the triboelectric/electrode materials? In Fig. 2i, the air gap was varied from 0 to 2 mm. Is 0 mm a contact-mode TENG?

Response: We appreciate the reviewer for raising this good question. In the manufacturing process, we used the acrylic sheet with different thicknesses as spacers to form an air gap between the rotor and stator, and the gap can be obtained precisely

by further considering the thickness of the triboelectric/electrode materials. It is worth noting that the air gap doesn't include the thickness of triboelectric/electrode materials. Here, five spacers (2 mm × 4 mm) are placed evenly on the edge of the stator to ensure that the rotor can run stably under a non-contact state. In **Fig. 2i**, the gap of 0 mm means the full-contact TENG. We have made some appropriate supplements in the article.

The methods section: (4) Assembly: Acrylic sheets (2 mm × 4 mm) with different thicknesses (0.35 mm, 0.65 mm, 1 mm, 1.5 mm, 2 mm) were used as spacers to form a certain gap between the stator and the slider to achieve a non-contact mode.

Fig. 2i The output charge of plane FSS-TENG with different electrode area and air gaps (the gap of 0 mm represents the full contact state).

Comment 3:

The authors state that “The surface of the electrodes is covered with a layer of 25 μm PA film”, but do not mention the process used in this covering step. PA should also be defined in the text. Why was PA chosen?

Response: We appreciate the reviewer's detailed reviewing and valuable comments. PA is a kind of electropositive polymer whose full name is Polyamide, which can be also called Nylon.

In the covering process, a very thin layer of Epoxy glue is coated on the surface of the Cu electrode by scraping film approach first, and then the PA film is covered on the epoxy glue on one side, and in the last the PA film is pasted on the surface of Cu electrode by epoxy glue. Especially, the thickness of the epoxy glue layer can be neglected due to its very thin thickness (less than 1 μm).

The dielectric film covered on the electrode surface can effectively inhibit air breakdown, so we chose three films with different electronegativity for investigation. In **Fig. 2j**, we explored the influence of PA/Kapton/PTFE dielectric film covered on the excitation electrode on the output performance. The charge on the PA film is easy to dissipate and does not accumulate a high charge on the film surface to shield the charge

on the electrode, while Kapton and PTFE electrets have strong electronegativity and a strong ability to retain electrons, which can easily form electrostatic shielding between induction electrodes. **Supplementary Figures 8 and 9** show the detailed dynamic excitation processes of TENG with the stator and slider electrodes attached to different dielectric films, showing that the PA film is the best choice.

We have made some modifications in the corresponding part of the article, and have also given the definitions of PA and PTFE.

The surface of the electrodes is covered with a layer of 25 μm nylon (PA) film, which can effectively inhibit the direct air breakdown between the metal electrodes, thus obtaining a larger charge density.

Because the charge on the PA film is easy to dissipate and is not accumulated on the film surface to shield the charge on the electrode, a stable and highly effective output is achieved. However, Kapton and PTFE are both electrets, they have strong electronegativity and a strong ability to hold electrons, which can easily form an electrostatic shielding between induction electrodes and decrease the output charge.

Comment 4:

What are the VMC components for the experiments performed in section 2.1? How does the overall performance change with these components?

Response: Thank the reviewer for the detailed questions. Our work used the second-order voltage-multiplying circuit (VMC), and the photograph is shown in **Supplementary Fig. 1**, and it consists of five diodes and five ceramic capacitors.

In 2019, our group reported the idea of self-charge excitation using VMC for the first time (*Nat. Commun.* **2019**, *10*, 1426). The self-charge excitation is realized by the auto switching of the external capacitors between series and parallel connections during sliding cycles. In recently reported work (*Adv. Energy Mater.* **2021**, 2100050), the authors have investigated the output performance of the different number of VMC unit (N) and the capacitance value of external capacitors for the contact-separation mode TENG. As shown in Figure R1(b, c), the inevitable energy loss and finite output of

TENG prevent the voltage across VMC from increasing linearly with the increase of N. When the supplementary charges and the leaking charges are balanced, the charge density will reach saturation. In Figure R1 (d), as the capacitance value of the VMC increases, the output charge density will also increase. Nevertheless, charge accumulation time shows an evident prolongation.

Figure R1. Output performance of TENG via fast charge accumulation. a) Simplified circuit diagram of self-charge excitation TENG with N-order VMC, in which N = 2, 3, 4, 5. b) Charge density and c) current density of TENG integrated with different VMC unit (dielectric layer: PVDF, operation frequency: 2.5 Hz). d) Charge accumulation time under different capacitance value of capacitors in circuit. (Ref. *Adv. Energy Mater.* **2021**, *2100050*) (for a response only). (<https://doi/10.1002/aenm.202100050>)

Different from the above self-charge excitation for contact-separation mode TENG, we use the VMC as a high-voltage source in our work, as shown in **Fig. 1c**, to make sure the charge on the excitation electrode reaches the maximum. At last, a new self-excited strategy is developed for sliding mode TENG. We also investigated the effect of different capacitors (0.47 nF, 1 nF, 2.2 nF, 3.3 nF, 4.7 nF) on the excitation voltage in the second-order VMC (N=2) as shown in **Supplementary Fig. 10**. The excitation voltage can reach about 2000 V. According to Paschen's law, the voltage that causes air breakdown between two parallel plates follows: $V_{a-b} = \frac{A(Pd)}{\ln(Pd)+B}$, so the voltage provided by the second-order VMC is sufficient, and chosen the small capacitors can

shorten the excitation time.

Supplementary Fig. 10 The excitation voltage of FSS-TENG under different capacitors.

The photograph of the VMC is shown in *Supplementary Fig. 1*, and it consists of five diodes and five ceramic capacitors.

In addition, the influence of the capacitance value in VMC on excitation charge is also tested⁴⁹. In order to have a faster excitation speed, five small capacitances are chosen for comparison as shown in *Supplementary Fig. 10*, from which we find that as the capacitance increases, the excitation time also increases. In sufficient time, the second-order VMC voltage with different capacitances can reach stable 2000 V, and this excitation voltage can enable the excitation electrode to reach the maximum charge. Therefore, in the whole testing process, the second-order VMC with 2.2 nF capacitance is used.

Comment 5:

How was Q_{\max} measured experimentally and how were the different gaps obtained and measured?

Response: Thank the reviewer for raising the detailed and valuable questions. We used

the plane structure to explore the maximum charge density (Q_{Max}) of non-contact TENG, and the triaxial displacement platform could accurately adjust the distance between the slider and the stator (the accuracy could reach 0.05mm). In order to measure Q_{Max} , we applied a high voltage source to generate a controllable voltage to TENG. The test schematic diagram is shown in the inset of **Fig. 2b**. In the sliding process, an adjustable positive high voltage is applied between the top and bottom electrodes, which can continuously supplement the charge to the electrode on the slider. By adjusting the value of voltage, the charge on the induction electrode under different gaps can reach saturation, that is, Q_{Max} can be measured.

Different gaps were achieved by setting acrylic spacers with different thicknesses between the stator and the rotor. Therefore, the gap can be measured by subtracting the thickness of the triboelectric/electrode layer from that of the spacer, and the thickness of the spacer is tested by a Digital Micrometer (the accuracy is 0.001mm).

Comment 6:

In Fig. S4 the excitation voltage seems to behave almost similarly already for 2 and 3 pairs. What is the reason for that?

Response: Thanks for the reviewer's detailed and careful review. **Supplementary Fig. 4b** shows the excitation voltage from the VMC at different pairs of sliders (a PTFE and a electrode in one pair). Commonly, the excitation voltage across the TENG also causes the leakage current of the dielectric layer. With the increase in slider pair number, a large dielectric layer area leads to a larger charge leakage which would result in a longer excitation time. On the other hand, the more induced charge can be supplied in one cycle with more pair numbers, which reduces the excitation time simultaneously. When the supplement charge and the leaking charge are in balance, the charge density and the excitation voltage reach the saturation.

Therefore, with increasing the slider pair number from 1 to 2, the charge increases by twice and the time taken to stability reduces by half (from 130 s to 68 s) at work frequency of 2 Hz . The excitation time cuts by one third (from 68 s to 48 s) when the

pair number increases from 2 to 3. Although the excitation voltage curve in the initial stage for 2 pairs seems to be almost similar to that for 3 pairs due to the slight difference of initial states (The initial state has a huge influence on the waveform of the output curve), the excitation time still becomes shorter with 3 pair sliders at last.

Moreover, as the frequency increases (the speed of excitation is faster), the excitation time can be further shortened. At 300 rpm, the excitation time to reach saturation is only 5 s (**Fig. 3 b-d**).

It is clear that the increase in the excited charge is proportional to the pairs, but the excitation voltage finally tends to the same value, because the value is determined by the order of VMC, and the pairs only affect the speed of voltage growth. When the slider pair (a PTFE and an electrode in one pair) increases from one to three, the more induced charge can be supplied in one cycle, by which the excitation voltage takes less time for 130 s, 68 s and 48 s to reach stability at a frequency of 2 Hz, respectively. Although the excitation voltage curve for 3 pairs seems to be almost similar for 2 pairs due to the slight difference of initial states (The initial state has huge influence on the waveform of output curve), the excitation time is still shorter with 3 electrodes at last.

Comment 7:

With increasing rotational speed, the transferred charge eventually starts to decrease. The authors claim that “higher rotational speeds may affect the induction between the electrodes”, but it is not clear how this can occur.

Response: We appreciate the reviewer for this suggestion. In the test process of the FSS-TENG, a higher rotating speed generates greater centrifugal motion and vertical tremor limited by the imperfect fabrication technology, which makes the gap between the rotator and the stator increase significantly under high speed, and results in the decrease of maximum output charge finally. We have added the explanations in the manuscript to clarify this point.

while higher rotational speed affects the gap between the electrodes. When the speed increases, the centrifugal force and vertical tremor of the rotator increase

correspondingly due to the imperfect fabrication craft, causing a increase in the gap between the rotor and the stator. So the maximum output charge declines with the increase in speed.

Comment 8:

The charging of the different capacitors using the self-exciting TENG is not compared with the same process with the non-exciting case. What is the charging improvement with the proposed concept?

Response: We appreciate the reviewer for this good question. We have tested the voltage-time curve of charging capacitors with the F-TENG at the same process and the results are shown in **Supplementary Fig. 12**. A detailed comparison of the charging process of FSS-TENG and F-TENG is listed in the table below. It can be seen that the charging rate can be improved by 3~4 times with the self-excited strategy. Notably, it will beyond the working voltage of the capacitor when charging a 100 μF capacitor by FSS-TENG at 300 rpm.

Supplementary Fig. 12 Voltage curves of capacitors charging by F-TENG at 300 rpm.

Time (s)		181	300			
Capacitor		100 μ F	220 μ F	470 μ F	1 mF	
Voltage (V)	FSS-TENG	50	Over range	31.3	16.5	5.43
	F-TENG	14.74	23.03	8.98	4.27	1.60

Comment 9:

The EMU presented in Fig. 5d should be clearly detailed.

Response: We thank the reviewer for this good suggestion. The energy management unit in **Fig. 5d** consists of a spark switch with a working voltage of 400 V, an inductance of 20 mH, a 110 pF input capacitor and a 470 μ F output capacitor. We also added this message to the manuscript.

With the help of the energy management unit (EMU), which consists of a spark switch with a working voltage of 400 V, an inductance of 20 mH, a input capacitor of 110 pF and a output capacitor of 470 μ F, as presented in **Fig. 5d**, the charging process for capacity by the FSS-TENG is greatly improved.

Comment 10:

Recent reviews on biomedical (Recent advances of triboelectric nanogenerator based applications in biomedical systems; <https://doi.org/10.1002/eom2.12049>), human-machine (Human–Machine Interfacing Enabled by Triboelectric Nanogenerators and Tribotronics; <https://doi.org/10.1002/admt.201800487>), high voltage (High-voltage applications of the triboelectric nanogenerator—Opportunities brought by the unique energy technology; <https://doi.org/10.1557/mre.2020.2>) and blue energy (Emerging triboelectric nanogenerators for ocean wave energy harvesting: state of the art and future perspectives; <https://doi.org/10.1039/D0EE01258K> and Triboelectric nanogenerators for a macro-scale blue energy harvesting and self-powered marine environmental monitoring system; <https://doi.org/10.1039/C9SE01184F>) triboelectric

nanogenerators have been published and should be cited in the manuscript.

Response: Thanks for the reviewer's good suggestions. We have cited the works mentioned above in suitable positions for the manuscript.

Comment 11:

There is a small number of typos in the manuscript that should be corrected (e.g. "Tow" in Fig. S4).

Response: Thanks for the reviewer's detailed reviewing. We have corrected the errors accordingly, and we also check and correct the whole manuscript carefully.

Reviewer #2 (Remarks to the Author):

This manuscript presents the floating self-excited sliding TENG (FSS-TENG) having high charge density based on air breakdown model. The non-contact TENG has been demonstrated to overcome the durability and efficiency issue of traditional TENG, however, charge density and output power should be improved. The author proposed the air breakdown model to calculate the optimal structure for higher charge density. It is remarkable that the FSS-TENG first achieved the first charge density of $71.53 \mu\text{C}/\text{m}^2$, which is 5 times compared to traditional floating TENG. However, the reviewer thinks this manuscript needs additional data supplement to be published in Nature Communications after minor revision. The required revision data are summarized as follows.

Response: We highly appreciate the reviewer's positive comments on our work. And we also thank the reviewer's detailed and responsible review of our work.

Comment 1:

Regarding the transfer charge (Fig. 2j), the material properties-based explanation is recommended for clear understanding. The manuscript compared theoretically for different electrode gap and area only. The reviewer suggest that the material analysis of Kapton, PA, and PTFE should be included.

Response: Thanks for the reviewer's good suggestion. The material property is an important factor affecting the output performance of TENG. In this work, the purposes of the dielectric film are to prevent the electric breakdown between the top and bottom electrodes to protect the device and to inhibit the leakage of charges on the electrode surface to obtain a larger output charge.

We chose three materials with different abilities for fettering charges (positive: PA> Kapton> PTFE: negative, triboelectric series: *Nat. Commun.* **2019** *10*, 1427), to explore the influence of different materials under self-exciting behavior. First, PA is not an electret material, so the positive charge on the PA's surface is easy to dissipate and can't accumulate to a high charge density (*Nat. Commun.* **2020** *11*, 6186), which can't shield the charge on the electrode and enables larger output charge. Second, Kapton and PTFE

are both electrets, they have strong electronegativity and a strong ability to retain electrons, which can easily form an electrostatic shielding between induction electrodes and decrease the output charge. **Supplementary Figures 8 and 9** show the detailed dynamic excitation processes of TENG with the stator and slider electrodes attached to different dielectric films. It can be found that the charge value and excitation speed are different in different cases (the initial charge are almost the same). Due to the property of charge dissipation, the output charge is the largest with PA film on the surface of both stator and slider electrodes. We have supplemented this in the manuscript.

To avoid air breakdown, covering a layer of the dielectric film on the surface of the top and bottom electrodes is an effective method. Three kinds of dielectric film (positive: PA> Kapton> PTFE: negative) with different abilities for fettering charge are chosen for the test ⁴⁸. **Figure 2j** displays the transfer charge with different materials under an air gap of 0.35 mm. It reveals that the transfer charge is largest with PA film on the surface of both stator and slider electrodes. Because the charge on the PA film is easy to dissipate and does not accumulate to shield the charge on the electrode, a stable and highly effective output is achieved. However, Kapton and PTFE are both electrets, they have strong electronegativity and a strong ability to hold electrons, which can easily form an electrostatic shielding between induction electrodes and decrease the output charge.

Comment 2:

In addition, the detail explanation should be required for comparing the charge depending on rpm and wind speed (Fig. 3e and 4e). Although the very high rpm affected the induction to lower the transferred charge, high wind speed didn't induce low charge.

Response: Thanks for the reviewer's good suggestion. **Fig. 3e** shows the output charge of TENG at different rotating speeds. In which, the output charge increases with the enhancement of rotating speed at first (**Output charges increase when less than 300 rpm**). However, higher rotating speed generates greater centrifugal motion and vertical

tremor limited by the imperfect fabrication craft, which makes the gap between the rotator and the stator increase significantly under high speed, and results in the decrease of maximum output charge finally (Output charge declines when speed is larger than 300 rpm). However, it can be seen in Fig. 4e that the output charge has no decrease trend at different wind speeds, mainly because the corresponding rotational speed is still relatively low even at high wind speed (19~140 rpm). Besides, the uneven force and large fluctuation of wind energy easily lead to large vertical tremors and **decrease the transfer charge at low speed**. Consequently, the output charge of FSS-TENG shows a constant tendency due to the offset of the above two factors when it is driven by wind at last. We have clarified this point in the manuscript. The rotational speed of the rotator at different wind speeds has been added in **Supplementary Table 2** as well.

Wind speed (m s ⁻¹)	3	4	5	6	6.5	7
rotational speed (rpm)	19.74	71.43	99.24	117.18	131.78	139.34

Supplementary Table 2. The rotational speed of the rotator at different wind speeds. At different wind speeds, the rotator speed is obtained by calculating the cycle number of measured transfer charge per second.

The results are plotted in Fig. 4e-g under different wind speeds from 3 to 7 m s⁻¹, from which we can see that the transferred charge remains about 420 nC, different from the trend driven by a motor, mainly because the corresponding rotational speed is still relatively low even at high wind speed (19~140 rpm). (*Supplementary Table 2*). Besides, the uneven force and large fluctuation of wind energy easily lead to large vertical tremors and decrease the transfer charge at low speed. Consequently, the output charge of FSS-TENG shows a constant tendency due to the offset of the above two factors when driven by wind at last.

Comment 3:

The quality of some figures should be improved for better understanding. So below papers are recommended to be referred to enhance the informativeness of figures.

- A. "Stretchable piezoelectric nanocomposite generator." Nano Convergence, 3, 12, 2016
- B. "Self-assembled incorporation of modulated block copolymer nanostructures in phase-change memory for switching power reduction." Acs Nano, 7(3), 2651, 2013

Response: Thanks for the reviewer's good suggestions. We have referenced the above works to enhance the informativeness of figures.

Reviewer #3 (Remarks to the Author):

The paper draft is interesting, and a novel device is proposed.

There are several notable grammatical issues that should be resolved prior to publication. Specifically, the article is missing the article “the” throughout and this oversight makes the draft difficult to read.

Response: We highly appreciate the reviewer’s positive comments on our work and also thank the reviewer’s detailed and responsible review of our work. We have examined the full text carefully and added the word ‘the’ in the corresponding position to make the draft read easily.

In addition, the authors give several statements such as, “ The air breakdown model of non-contact TENG is first given to achieve a maximum charge density output,” where the article is missing and the way in which “first” is used gives the implication that there is a second item to be described in the section or paragraph. However, the reviewer believes the authors intend these sentences to reflect that this work/device is described “for the first time”. Hence, sentences like this in the draft should be modified for clarity.

Response: Thanks for the reviewer’s detailed and good comments. We have corrected the errors in the whole manuscript accordingly.

There are also a few instances in the Introduction that could be improved. For example, for the sentence that begins with “Based on the” in lines 51 – 55, the authors state that the TENG is more efficient than an electromagnetic generator and piezoelectric generator, however, no actual numbers are provided for comparison. Indeed, the authors do provide two references for this sentence. However, the provision of actual numbers in the parenthesis in this sentence would strengthen the claim.

Response: Thank the reviewer for this good suggestion. In the introduction part, we have provided actual numbers in some sentences for clear comparison to outstand the arguments. And the revised texts are listed below.

triboelectric nanogenerator (TENG) has been demonstrated as a more efficient energy harvesting strategy, compared with electromagnetic generator^{7,8} or piezoelectric generator⁹⁻¹¹ for low-frequency ambient distributed energy (5~100 times smaller than the TENG at < 5 Hz working frequency).

In sentence, 61, the authors state that “contact has high durability and almost 100% high conversion efficiency.” This claim would be enhanced if the authors report the actual value versus a rounded number as it is clear that 100% efficiency is not possible.

Response: Thank the reviewer for the good comments. Previous work reported that non-contact TENG would have a 100% theoretical conversion efficiency in an ideal situation (*Adv. Mater.* **2014**, *26*, 2818–2824) and the actual conversion efficiency would reach 85% (*Adv. Mater.* **2014**, *26*, 6599–6607). Ideally, if there is not any friction in the environment, the theoretical total conversion efficiency of the non-contact mode is expected to be 100%. However, in the real situation, air friction, a non-neglectable factor, brings about an energy loss for the device working against it. Therefore, considering various energy consumption, the conversion efficiency of non-contact TENG should reach 85%. We are sorry that this point confuses the reviewer, and we have added some descriptions in the manuscript to declare the working parameters in our case.

a floating mode TENG without interface contact has high durability and would have almost 100% theoretical conversion efficiency owing to the zero interfacial friction loss²⁹

The sentence on line 167 is missing a verb.

Response: Thanks for the reviewer’s detailed reviewing. We have corrected the errors accordingly.

For the non-contact sliding TENG, the equivalent physical model is shown in **Fig. 2a**.

The authors provide an equation for voltage between two parallel plates and maximum charge density that includes variables, P, A, and B, which are functions of composition and pressure of the air. The authors assume and validate this model using COMSOL assuming that the air is under these conditions. However, wind turbines are many meters above sea level and therefore, the air has different characteristics than those assumed in the model. This paper would be enhanced if the theoretical and numerical models took more realistic parameters for this application into account. In doing this, others using the results would have a better understanding of the advantages and limitations of the device.

Response: Thank the reviewer for the insightful comment. On the one hand, environmental factors like high humidity could decrease the output charge of TENG significantly, so encapsulation is usually necessary for TENG to obtain stable and reliable output in different application scenes, which can efficiently avoid kinds of negative environmental impacts. On the other hand, the keynotes of this work are the novel floating self-excited sliding TENG (FSS-TENG) and its unique output performance. This FSS-TENG can efficiently harvest kinds of micro mechanical energy likes wave energy, human motion etc., and the wind energy harvesting is just one applied demonstration. Air factor is not the most important focus, and so we don't discuss the influence of air on wind turbines in depth from theoretical and numerical models.

In the Optimization of Structural parameters section, the authors describe how the various parameters such as the number of electrodes are varied, but an approach to the optimization strategy, approach, or analytical model associated with the things such as the number of electrodes is not described in a meaningful way. Could the approach to the optimization methods be described in a generalized fashion? Can this approach be mathematically modeled?

Response: Thank the reviewer for raising this detailed and valuable question.

First, for the traditional sliding generator, the optimization of structural parameters

like electrode gap, pairs, freestanding height etc. to obtain maximum output power **has been systematically explored from experiment and mathematics** (*Nano Energy*, **2015**, 12 760–774). The FSS-TENG belongs to a kind of sliding mode TENG, so the above conclusion is also suitable for the FSS-TENG. However, **the most key point that affects the output performance of TENG is the total output charge and charge density** (*Nat. Commun.* **2015** 6, 8376). So enhancing the output charge density of non-contact sliding TENG is our focus.

Second, to maximize the output charge of non-contact sliding TENG by self-excited strategy, we discuss its impact factors on output charge density from experiment and theory. One is the excitation voltage which is proportional to the charge density (Equation 1), so high excitation voltage is needed to obtain large charge density.

$$\sigma = \frac{\varepsilon_0}{d} \cdot V_E \quad (1)$$

Another is the limitation of air breakdown, air breakdown effect leads to the existence of a maximum charge density for a TENG (Equation 6), and theoretical and experimental results both verify that a small air gap contributes to larger charge density.

$$\sigma_{Max} \leq \frac{AP\varepsilon_0}{\ln(Pd)+B} \quad (6)$$

Therefore, a high excitation voltage and small air gap distance are necessary to gain a large charge density.

Third, the relationship of total output charge Q , charge density σ , area of one pair electrode S_s and slider pairs (a PTFE and a electrode in one pair) K are given below.

$$Q = \sigma \cdot S_s \cdot K$$

From the equation above, we can see that large charge density, electrode area and more pairs all help to achieve large output charge, and this is also verified in **Fig. 2i** and **Fig. S4**.

In summary, we also discuss the maximum output charge through structure optimization from the mathematical models and experimental demonstration.

REVIEWER COMMENTS

Reviewer #1 (Remarks to the Author):

The revised version of the manuscript answered all the raised questions satisfactorily and is therefore ready to be published.

Reviewer #2 (Remarks to the Author):

This manuscript argues for FSS-TENG by leading to improvement on non-contact TENGs with existing deficient outputs. It shows a very superior performance in endurance tests with TENG compared with conventional contact mode and shows a higher output than in previous non-contact mode TENGs. It is true that it represents superior outputs and proves their interest, but by contrast, the part about the application and the structure of the paper seems to require a lot of addition. For the following reasons, this paper would like to give a revision.

What was noted in the previous review, 1. Material analysis of Kapton, PA, and PTFE was conducted (Figure 2.j) 2. The relationship between wind speed and RPM was also added in the text. Author explained in manuscript that the rpm is low despite the high wind speed. 3. The quality of explanations and figures has been found to have similar papers, as in No. 1 of the second review, and it still needs to be improved.

Also, after reviewed the revised manuscript and had additional points such as #2, #3, and #4 of the second review.

1. The structure of the paper and its explanation are very similar to the " Integrated charge excitation triboelectric nanogenerator " published in previous nature communication (10, 1426, 2019). It is thought that the quality of the pictures and graphs needs to be improved. Furthermore, in Figure 1(b), the difficulty of matching the colors of each materials seems to need to be improved.

2. Turning on 912 LEDs is suitable to show the output of the device. However, it seems necessary to modify the experimental setting. In the text, experiments or applications were conducted at wind speeds of more than 3 m/s. The average annual wind speed is less than 2 m/s, and the actual application is expected to be difficult. To claim the application and output shown in the text, it would be necessary to show the operation and application of the device at 1 m/s. In the text, author claim 'We believe that the FSS-TENG could be driven easily at less than 1 m s⁻¹ wind speed with the further optimization of the manufacturing process.' If the device operation at 1 m/s is not shown, the application claimed in the text is unlikely to be meaningful, and the application field must be changed.

3. When used outdoors, simple experiments and data are needed to ensure that there is no degradation of ultraviolet rays, temperature changes, humidity, and sea breeze.

4. For transparency of experiments, detailed descriptions and illustrations of the manufacturing process are required.

Over all, three papers are recommended to be referred to enhance the informativeness of figures and application.

A. Achieving high-resolution pressure mapping via flexible GaN/ZnO nanowire LEDs array by piezophototronic effect, *Nano Energy*, 58, 633, 2019

B. Wireless smart contact lens for diabetic diagnosis and therapy, *Science Advanced*, 6, 17, 2020

C. Piezoelectric Energy Harvesting from Two-Dimensional Boron Nitride Nanoflakes, 11, 37920, 2019

Point-by-Point Response to the Reviewer's Comments
(Comments in black, response in blue)

Reviewer #1 (Remarks to the Author):

The revised version of the manuscript answered all the raised questions satisfactorily and is therefore ready to be published.

Response: Thanks for your positive comments and the approval on our work.

Reviewer #2 (Remarks to the Author):

This manuscript argues for FSS-TENG by leading to improvement on non-contact TENGs with existing deficient outputs. It shows a very superior performance in endurance tests with TENG compared with conventional contact mode and shows a higher output than in previous non-contact mode TENGs. It is true that it represents superior outputs and proves their interest, but by contrast, the part about the application and the structure of the paper seems to require a lot of addition. For the following reasons, this paper would like to give a revision.

Response: We appreciate the reviewer's positive comments on our work. And we also thank the reviewer's detailed and responsible review of our work.

What was noted in the previous review, 1. Material analysis of Kapton, PA, and PTFE was conducted (Figure 2.j) 2. The relationship between wind speed and RPM was also added in the text. Author explained in manuscript that the rpm is low despite the high wind speed. 3. The quality of explanations and figures has been found to have similar papers, as in No. 1 of the second review, and it still needs to be improved.

Also, after reviewed the revised manuscript and had additional points such as #2, #3, and #4 of the second review.

Comment 1:

The structure of the paper and its explanation are very similar to the " Integrated charge excitation triboelectric nanogenerator " published in previous nature communication (10, 1426, 2019). It is thought that the quality of the pictures and graphs needs to be

improved. Furthermore, in Figure 1(b), the difficulty of matching the colors of each materials seems to need to be improved.

Response: Thanks for the reviewer's detailed comments.

We greatly respect the comments from the reviewer, however, the highlights of this work are completely different from that of our pervious work entitled "Integrated charge excitation triboelectric nanogenerator". In pervious work, we systematically develop external charge excitation and self-charge excitation strategies for effectively boosting the output charge of contact-separation mode TENG. Especially, for the self-charge excitation TENG, the automatic serial-parrallel switch of capacitors in SVMC during the contact-separation process of TENG, causing the self-increase of output charge in TENG (Fig. R1). In this work, we first developed a self-excited amplification method for effektivly enhancing the output of non-contact sliding mode TENG, in which the positive feedback between the slider electrode and stator electrode leads to the continuous output charge self-increase (Fig. 1) (Stator electrode can excite charge into slider electrode through the VMC, increasing charge in slider electrode which boots the output charge of stator electode by electrostatic induction, and then more charge is excited into sldier electrode, thus a positive feedback forms). Obviously, both the working principle and applied target of above two works are completely different.

We have improved the quality of pictures and graphs, and in Figure 1(b) we have used distinct colors to represent different materials.

Fig. R1 Principle of the self-charge excitation strategy for contact-separation mode triboelectric nanogenerator. (*Nat. Commun.* 2019, 10, 1426)

Figure 1. Structure and working principle of the floating self-excited sliding TENG

(FSS-TENG).

Comment 2:

Turning on 912 LEDs is suitable to show the output of the device. However, it seems necessary to modify the experimental setting. In the text, experiments or applications were conducted at wind speeds of more than 3 m/s. The average annual wind speed is less than 2 m/s, and the actual application is expected to be difficult. To claim the application and output shown in the text, it would be necessary to show the operation and application of the device at 1 m/s. In the text, author claim 'We believe that the FSS-TENG could be driven easily at less than 1 m s⁻¹ wind speed with the further optimization of the manufacturing process.' If the device operation at 1 m/s is not shown, the application claimed in the text is unlikely to be meaningful, and the application field must be changed.

Response: Thank the reviewer for raising up this question.

Firstly, According to the classification of wind power, the wind speed from 3.4 m/s to 5.4 m/s are breeze, which can make the red flag unfold. In this work, FSS-TENG can be driven at a wind speed of 3 m/s. Moreover, in the description of **Figure 1a**, our application blueprint is to place FSS-TENG on **coastal areas and mountainous areas** to collect wind energy to realize distributed energy collection. The airflow at the top of the mountain or on the coast is relatively strong, and **the wind speed is usually larger than 3 m/s**, enough to drive the FSS-TENG, so we cannot use the average annual wind speed to judge its actual application.

Secondly, The statement 'We believe that the FSS-TENG could be driven easily at less than 1 m s⁻¹ wind speed with the further optimization of the manufacturing process.' is our outlook. However, due to the limitation of the manufacturing process from available experiment materials and tools, we have to use the acrylic sheet as **spacers** on the edge of the stator to form an air gap between the rotor and stator. **Which results in the running resistance of the device**, so our self-made FSS-TENG can only be driven at

a wind speed of 3 m/s. Actually, the spacer can be removed by adopting advanced manufacturing process, and to make the device can be driven by wind speed below 3 m/s, which is belong to an engineering problem and is not our main focus. Therefore, we proposed the above rational outlook. When the rotor and the stator are separated by 3 cm (there is no the running resistance between the spacers and the rotor), and FSS-TENG can operate at wind speed less than 1m/s, as shown in Fig. R2 . We also provide the corresponding video.

Thirdly, in this work, we demonstrated the application of self-excited amplification method on non-contact sliding TENG and gave out the air breakdown model of non-contact TENG for the first time. The above two points are our highlights and focuses.

Fig. R2 The picture of FSS-TENG being driven by wind speeds below 1m/s. The rotor and the stator are separated by 3 cm.

Comment 3:

When used outdoors, simple experiments and data are needed to ensure that there is no degradation of ultraviolet rays, temperature changes, humidity, and sea breeze.

Response: Thank the reviewer for the detailed comment. The output stability of TENG will definitely be affected by various external environments, such as temperature changes and high humidity, etc.

As shown in **Supplementary Fig. 11**, the results displays that the change of humidity (from 35 to 85%RH) and temperature (from 15°C to 60°C) decreases the output of FSS-TENG by value of 14% and 26%, respectively. As the device expects to work in outside door, we also detect the influence from solar light under the irradiation of the simulated sunlight (containing ultraviolet rays), as shown in **Supplementary Fig. 12**. When the FSS-TENG works under the illumination with the simulated solar light in the standard light intensity of 100 mW/cm² for a half hour, the output charge of FSS-TENG decreases slightly, which is mainly caused by the increase of device temperature (**Supplementary Fig. 12**). Thus, the environmental ultraviolet ray (cool light) has a little influence on FSS-TENG's output charge. And we are so sorry that there is no the experimental conditions to measure the influence of sea breeze since the the location of our lab is far from the sea.

Besides, the humidity and ultraviolet rays can be effectively avoided by encapsulating the device. In the Methods Section of the manuscript, we also explained the data testing environment, 'The temperature was controlled within 15-35°C and humidity about 35-55% RH. ' It also shows that the daily temperature changes have little effect on the output of FSS-TENG.

Supplementary Fig. 11. The surface charge density of FSS-TENG under different temperature **a** and humidity **b**.

Supplementary Fig. 12 The surface charge density of FSS-TENG under the simulated solar light. The standard light intensity is 100 mW/cm². The test humidity is controlled at around 50%.

The effect of different temperature and humidity on the surface charge density of FSS-TENG are shown in **Supplementary Fig. 11**. The output shows a slight downward trend with the increase of temperature and humidity. Because the electron thermionic emission effect of induced charges on the surface of the electrode increases when the temperature rises, and more water molecules take away the charges on the electrode surface in high-humidity air, which reduces charge density and output performance of FSS-TENG. When the FSS-TENG works under the illumination with the simulated solar light in the standard light intensity of 100 mW/cm² for a half hour, the output charge of FSS-TENG decreases slightly, which is mainly caused by the increase of device temperature during the irradiation process (**Supplementary Fig. 12**). So the temperature is controlled within 15-35°C and humidity about 35-55% RH of the experimental test environment.

Comment 4: For transparency of experiments, detailed descriptions and illustrations of the manufacturing process are required.

Over all, three papers are recommended to be referred to enhance the informativeness of figures and application.

A. Achieving high-resolution pressure mapping via flexible GaN/ZnO nanowire LEDs array by piezo-phototronic effect, Nano Energy, 58, 633, 2019

B. Wireless smart contact lens for diabetic diagnosis and therapy, Science Advanced, 6, 17, 2020

C. Piezoelectric Energy Harvesting from Two-Dimensional Boron Nitride Nanoflakes, 11, 37920, 2019

Response: Thanks for the reviewer's detailed suggestions. Detailed manufacturing processes are shown in the Method part. We have referenced the above works to enhance the informativeness of figures.

Point-by-Point Response to the Reviewer's Comments
(Comments in black, response in blue)

Reviewer #1 (Remarks to the Author):

The revised version of the manuscript answered all the raised questions satisfactorily and is therefore ready to be published.

Response: Thanks for your positive comments and the approval on our work.

Reviewer #2 (Remarks to the Author):

This manuscript argues for FSS-TENG by leading to improvement on non-contact TENGs with existing deficient outputs. It shows a very superior performance in endurance tests with TENG compared with conventional contact mode and shows a higher output than in previous non-contact mode TENGs. It is true that it represents superior outputs and proves their interest, but by contrast, the part about the application and the structure of the paper seems to require a lot of addition. For the following reasons, this paper would like to give a revision.

Response: We appreciate the reviewer's positive comments on our work. And we also thank the reviewer's detailed and responsible review of our work.

What was noted in the previous review, 1. Material analysis of Kapton, PA, and PTFE was conducted (Figure 2.j) 2. The relationship between wind speed and RPM was also added in the text. Author explained in manuscript that the rpm is low despite the high wind speed. 3. The quality of explanations and figures has been found to have similar papers, as in No. 1 of the second review, and it still needs to be improved.

Also, after reviewed the revised manuscript and had additional points such as #2, #3, and #4 of the second review.

Comment 1:

The structure of the paper and its explanation are very similar to the " Integrated charge excitation triboelectric nanogenerator " published in previous nature communication (10, 1426, 2019). It is thought that the quality of the pictures and graphs needs to be

improved. Furthermore, in Figure 1(b), the difficulty of matching the colors of each materials seems to need to be improved.

Response: Thanks for the reviewer's detailed comments.

We greatly respect the comments from the reviewer, however, the highlights of this work are completely different from that of our pervious work entitled "Integrated charge excitation triboelectric nanogenerator". In pervious work, we systematically develop external charge excitation and self-charge excitation strategies for effectively boosting the output charge of contact-separation mode TENG. Especially, for the self-charge excitation TENG, the automatic serial-parrallel switch of capacitors in SVMC during the contact-separation process of TENG, causing the self-increase of output charge in TENG (Fig. R1). In this work, we first developed a self-excited amplification method for effektivly enhancing the output of non-contact sliding mode TENG, in which the positive feedback between the slider electrode and stator electrode leads to the continuous output charge self-increase (Fig. 1) (Stator electrode can excite charge into slider electrode through the VMC, increasing charge in slider electrode which boots the output charge of stator electode by electrostatic induction, and then more charge is excited into sldier electrode, thus a positive feedback forms). Obviously, both the working principle and applied target of above two works are completely different.

We have improved the quality of pictures and graphs, and in Figure 1(b) we have used distinct colors to represent different materials.

Fig. R1 Principle of the self-charge excitation strategy for **contact-separation mode** triboelectric nanogenerator. (*Nat. Commun.* **2019**, 10, 1426)

Figure 1. Structure and working principle of the floating self-excited sliding TENG

(FSS-TENG).

Comment 2:

Turning on 912 LEDs is suitable to show the output of the device. However, it seems necessary to modify the experimental setting. In the text, experiments or applications were conducted at wind speeds of more than 3 m/s. The average annual wind speed is less than 2 m/s, and the actual application is expected to be difficult. To claim the application and output shown in the text, it would be necessary to show the operation and application of the device at 1 m/s. In the text, author claim 'We believe that the FSS-TENG could be driven easily at less than 1 m s⁻¹ wind speed with the further optimization of the manufacturing process.' If the device operation at 1 m/s is not shown, the application claimed in the text is unlikely to be meaningful, and the application field must be changed.

Response: Thank the reviewer for raising up this question.

Firstly, According to the classification of wind power, the wind speed from 3.4 m/s to 5.4 m/s are breeze, which can make the red flag unfold. In this work, FSS-TENG can be driven at a wind speed of 3 m/s. Moreover, in the description of **Figure 1a**, our application blueprint is to place FSS-TENG on **coastal areas and mountainous areas** to collect wind energy to realize distributed energy collection. The airflow at the top of the mountain or on the coast is relatively strong, and **the wind speed is usually larger than 3 m/s**, enough to drive the FSS-TENG, so we cannot use the average annual wind speed to judge its actual application.

Secondly, The statement 'We believe that the FSS-TENG could be driven easily at less than 1 m s⁻¹ wind speed with the further optimization of the manufacturing process.' is our outlook. However, due to the limitation of the manufacturing process from available experiment materials and tools, we have to use the acrylic sheet as **spacers** on the edge of the stator to form an air gap between the rotor and stator. **Which results in the running resistance of the device**, so our self-made FSS-TENG can only be driven at

a wind speed of 3 m/s. Actually, the spacer can be removed by adopting advanced manufacturing process, and to make the device can be driven by wind speed below 3 m/s, which is belong to an engineering problem and is not our main focus. Therefore, we proposed the above rational outlook. When the rotor and the stator are separated by 3 cm (there is no the running resistance between the spacers and the rotor), and FSS-TENG can operate at wind speed less than 1m/s, as shown in Fig. R2 . We also provide the corresponding video.

Thirdly, in this work, we demonstrated the application of self-excited amplification method on non-contact sliding TENG and gave out the air breakdown model of non-contact TENG for the first time. The above two points are our highlights and focuses.

Fig. R2 The picture of FSS-TENG being driven by wind speeds below 1m/s. The rotor and the stator are separated by 3 cm.

Comment 3:

When used outdoors, simple experiments and data are needed to ensure that there is no degradation of ultraviolet rays, temperature changes, humidity, and sea breeze.

Response: Thank the reviewer for the detailed comment. The output stability of TENG will definitely be affected by various external environments, such as temperature changes and high humidity, etc.

As shown in **Supplementary Fig. 11**, the results displays that the change of humidity (from 35 to 85%RH) and temperature (from 15°C to 60°C) decreases the output of FSS-TENG by value of 14% and 26%, respectively. As the device expects to work in outside door, we also detect the influence from solar light under the irradiation of the simulated sunlight (containing ultraviolet rays), as shown in **Supplementary Fig. 12**. When the FSS-TENG works under the illumination with the simulated solar light in the standard light intensity of 100 mW/cm² for a half hour, the output charge of FSS-TENG decreases slightly, which is mainly caused by the increase of device temperature (**Supplementary Fig. 12**). Thus, the environmental ultraviolet ray (cool light) has a little influence on FSS-TENG's output charge. And we are so sorry that there is no the experimental conditions to measure the influence of sea breeze since the the location of our lab is far from the sea.

Besides, the humidity and ultraviolet rays can be effectively avoided by encapsulating the device. In the Methods Section of the manuscript, we also explained the data testing environment, 'The temperature was controlled within 15-35°C and humidity about 35-55% RH. ' It also shows that the daily temperature changes have little effect on the output of FSS-TENG.

Supplementary Fig. 11. The surface charge density of FSS-TENG under different temperature **a** and humidity **b**.

Supplementary Fig. 12 The surface charge density of FSS-TENG under the simulated solar light. The standard light intensity is 100 mW/cm². The test humidity is controlled at around 50%.

The effect of different temperature and humidity on the surface charge density of FSS-TENG are shown in **Supplementary Fig. 11**. The output shows a slight downward trend with the increase of temperature and humidity. Because the electron thermionic emission effect of induced charges on the surface of the electrode increases when the temperature rises, and more water molecules take away the charges on the electrode surface in high-humidity air, which reduces charge density and output performance of FSS-TENG. When the FSS-TENG works under the illumination with the simulated solar light in the standard light intensity of 100 mW/cm² for a half hour, the output charge of FSS-TENG decreases slightly, which is mainly caused by the increase of device temperature during the irradiation process (**Supplementary Fig. 12**). So the temperature is controlled within 15-35°C and humidity about 35-55% RH of the experimental test environment.

Comment 4: For transparency of experiments, detailed descriptions and illustrations of the manufacturing process are required.

Over all, three papers are recommended to be referred to enhance the informativeness of figures and application.

A. Achieving high-resolution pressure mapping via flexible GaN/ZnO nanowire LEDs array by piezo-phototronic effect, *Nano Energy*, 58, 633, 2019

B. Wireless smart contact lens for diabetic diagnosis and therapy, *Science Advanced*, 6, 17, 2020

C. Piezoelectric Energy Harvesting from Two-Dimensional Boron Nitride Nanoflakes, 11, 37920, 2019

Response: Thanks for the reviewer's detailed suggestions. Detailed manufacturing processes are shown in the Method part. We have referenced the above works to enhance the informativeness of figures.